# Demystifying and Enhancing the Efficiency of Large Language Model Based Search Agents

**Tiannuo Yang**[1,2*]**, Zebin Yao**[2*]**, Bowen Jin**[3]**, Lixiao Cui**[2]**, Yusen Li**[2]**, Gang Wang**[2]**, Xiaoguang Liu**[2]**, Willie Neiswanger**[1]

[1]University of Southern California  [2]Nankai University  [3]University of Illinois Urbana-Champaign

## Abstract

Large Language Model (LLM)-based search agents have shown remarkable capabilities in solving complex tasks by dynamically decomposing problems and addressing them through interleaved reasoning and retrieval. However, this interleaved paradigm introduces substantial efficiency bottlenecks. First, we observe that both highly accurate and overly approximate retrieval methods degrade system efficiency: exact search incurs significant retrieval overhead, while coarse retrieval requires additional reasoning steps during generation. Second, we identify inefficiencies in system design, including improper scheduling and frequent retrieval-induced stalls, which lead to cascading latency—where even minor delays in retrieval amplify end-to-end inference time. To address these challenges, we introduce `SearchAgent-X`, a high-efficiency inference framework for LLM-based search agents. `SearchAgent-X` leverages high-recall approximate retrieval and incorporates two key techniques: priority-aware scheduling and non-stall retrieval. Extensive experiments demonstrate that `SearchAgent-X` consistently outperforms state-of-the-art systems such as vLLM and HNSW-based retrieval across diverse tasks, achieving up to $3.4\times$ higher throughput and $5\times$ lower latency, without compromising generation quality. Code is available at https://github.com/tiannuo-yang/SearchAgent-X.

## 1 Introduction

Traditional Retrieval-Augmented Generation (RAG) typically uses a sequential retrieve-then-generate approach (Fan et al., 2024; Gao et al., 2024; Gupta et al., 2024; Huang & Huang, 2024; Wu et al., 2024; Yu et al., 2024; Zhao et al., 2024a;b), which limits dynamic interaction with knowledge bases. Recent advancements have ushered in RAG 2.0, known as *Search Agents* (Trivedi et al., 2022; Singh et al., 2025; Li et al., 2025; Jin et al., 2025a; Chen et al., 2025; Song et al., 2025; xAI, 2025; OpenAI, 2025). This paradigm leverages the strong reasoning capabilities of Large Language Models (LLMs), allowing for the dynamic and adaptive interleaving of reasoning steps with retrieval calls throughout the generation process. Instead of a fixed pipeline, search agents can decide *when* and *what* to retrieve based on LLM's ongoing reasoning, leading to significant improvements in the quality and depth of the generated responses. Leveraging post-training techniques similar to DeepSeek-R1, some pioneering models can even autonomously initiate retrieval actions during reasoning without intermediate supervision (Jin et al., 2025a; Chen et al., 2025; Song et al., 2025).

However, the improved generation quality achieved by search agents often comes at the cost of efficiency—an overhead that is nontrivial in practical deployments. In reasoning-with-search scenarios, achieving low-latency responses is critical for ensuring a seamless user experience (Ray et al., 2024; Jin et al., 2024). Moreover, during post-training of LLM-based search agents, efficient model rollouts over large-scale training corpora are essential to support scalable learning. While recent systems incorporate advanced inference optimizations—such as sequence concatenation (Jin et al., 2025a; Chen et al., 2025; Song et al., 2025) and prefix caching (Jin et al., 2024; Kwon et al., 2023; Zheng

---

[*]Equal contribution

Corresponding authors: Tiannuo Yang (`tiannuoy@usc.edu`), Lixiao Cui (`cuilx@nankai.edu.cn`), and Willie Neiswanger (`neiswang@usc.edu`).

et al., 2024)—these techniques are not specifically designed to address the unique computational challenges posed by the tight interleaving of multi-step reasoning and dynamic retrieval.

To this end, we first conduct a systematic analysis of the efficiency factors governing LLM-based search agents, uncovering insights that diverge from the understanding of naive RAG. Our in-depth analysis reveals two key observations: First, we demonstrate a non-monotonic relationship between retrieval accuracy and end-to-end efficiency. Both excessively high (e.g., exact search) and excessively low retrieval recall degrade overall efficiency. While aiming for perfect recall incurs unnecessary computational overhead in the retrieval phase, low recall necessitates more retrieval iterations and longer reasoning paths by the LLM to compensate (as shown in Figure 1). This highlights that search agent systems benefit from high-recall approximate search that effectively supports reasoning without unnecessary retrieval costs. Second, we find that search agent systems are highly sensitive to retrieval latency. Unlike naive RAG where retrieval is largely amortized, even minor increases in retrieval time in the search agent system can cause a disproportionately large increase in end-to-end latency (Figure 2a). We attribute this magnification effect to two primary root causes: *improper scheduling*, where standard policies like FCFS fail to prioritize requests that would benefit most from KV-cache reuse (Figure 2b), and *retrieval-induced stalls*, where timing misalignments between asynchronous retrieval and token generation force requests to wait, leading to unnecessary recomputation (Figure 2c).

Motivated by these findings, we propose `SearchAgent-X`, an inference system dedicated for efficient search agents. `SearchAgent-X` is designed to optimize end-to-end system throughput and latency by smoothly coordinating the interleaving of self-reasoning and retrieval. Since both overly low and high retrieval efforts lead to degraded efficiency, `SearchAgent-X` chooses to build upon a high-recall approximate retrieval method. To tackle the problem of improper scheduling, `SearchAgent-X` schedules requests with priority awareness through their real-time status to enhance KV-cache utilization. Moreover, in order to overcome frequent retrieval-induced stalls, `SearchAgent-X` proposes a non-stall retrieval mechanism through an adaptive strategy that allows generation to proceed without unnecessary waiting while ensuring sufficient retrieval quality.

Our extensive experiments demonstrate that `SearchAgent-X` consistently and significantly outperforms state-of-the-art baseline systems across various operational settings. In both offline and online inference scenarios, `SearchAgent-X` achieves substantial improvements in system performance (e.g., 1.3-3.4× higher throughput) by improving LLM KV-cache utilization (from 0.07 to 0.65), all while maintaining the high generation quality characteristic of search agents with exact retrieval.

## 2    BACKGROUND AND MOTIVATION

### 2.1    PRELIMINARY: LLM-BASED SEARCH AGENT SYSTEMS

LLM-based search agent systems are designed to tackle complex requests by decomposing problems into a series of interleaved, multi-turn reasoning and information retrieval steps. This allows the LLM to adaptively seek and integrate external knowledge throughout its reasoning process. Appendix A shows an example of the process of a LLM-based search agent.

**Supporting Multi-Turn Reasoning.** Search agent systems often build on LLM inference frameworks like vLLM (Jin et al., 2025a). They use *Sequence Concatenation* for dynamic retrieval: during inference, the system monitors model output for retrieval signals. Upon such a signal, LLM decoding pauses, a query is issued, and retrieved results are concatenated with previously generated tokens to form a new, extended *Sequence*. This is then re-injected into the LLM to resume reasoning.

To enhance efficiency, *Prefix Cache* is commonly leveraged (Jin et al., 2024; Zheng et al., 2024). This technique stores key-value (KV) pairs from the LLM's attention mechanism for prior tokens, allowing efficient reuse in subsequent generations. This is particularly advantageous in search agents, as the concatenated sequence's prefix, excluding newly retrieved tokens, overlaps with the previous generation. Furthermore, shared system prompts across search agent requests can be cached and reused. In our evaluation, enabling prefix caching saved over 24% of token recomputation costs.

**Sequence Scheduling.** Efficient scheduling is vital for high throughput. Modern LLM inference frameworks utilize *Iteration-Level Scheduling*, where GPU scheduling decisions occur at the granularity of the single token generation step (Kwon et al., 2023; Zheng et al., 2024). Compared to

sequence-level scheduling (NVIDIA, 2019; 2021), iteration-level scheduling avoids waiting for all sequences in a batch to complete, thus preventing bubble problems and becoming a leading solution. Frameworks like vLLM typically employ a First-Come-First-Serve (FCFS) scheduling policy.

**Retrieval Mechanism.** On the retrieval side, semantic search techniques efficiently locate relevant external knowledge. Queries are usually encoded into dense vector representations for searching in vector space. The two primary approaches are exact nearest neighbor (ENN) search (Dasgupta & Sinha, 2013) and approximate nearest neighbor (ANN) search (Malkov & Yashunin, 2018; Guo et al., 2020). Graph-based ANN methods like HNSW (Malkov & Yashunin, 2018) offer a favorable speed/accuracy trade-off, making them suitable for large knowledge bases.

## 2.2 KEY INSIGHTS: FACTORS GOVERNING EFFICIENCY

Despite significant progress in high-performance LLM inference and retrieval, the LLM-based search agent's efficiency remains poorly understood. In this section, we analyze the influence of two key factors: 1) retrieval accuracy and 2) retrieval latency, and examine how they contribute to severe inefficiencies in current solutions. For retrieval, we assume a local search with a *fixed dense encoder*. Experiments in this section use Search-R1 7B on MuSiQue and the default vLLM configuration.

### 2.2.1 IMPACT OF RETRIEVAL ACCURACY

**Insight 1:** *Both overly high and overly low retrieval recall degrade end-to-end efficiency. High recall increases retrieval overhead, while low recall leads to longer reasoning steps.*

We first investigate the impact of different retrieval accuracies on the system efficiency of search agents. Intuitively, lower retrieval accuracy means lower retrieval overhead, thus higher system efficiency. However, we observe a "less is more" phenomenon for LLM-based search agents. Low-recall retrievals may result in suboptimal context, forcing the model to compensate by issuing additional retrievals and extending the reasoning length. Figure 1 shows how varying the ANN search range affects throughput and average retrieval counts. When the search range is too small (e.g., 10), the model fails to retrieve useful documents, resulting in longer reasoning steps and an average of 6.5 retrievals per request. This reduces throughput to just above 2.1. As the search range increases to 500, retrieval quality improves, and the model completes reasoning with fewer retrievals (around 5.7), boosting throughput to over 3.2.

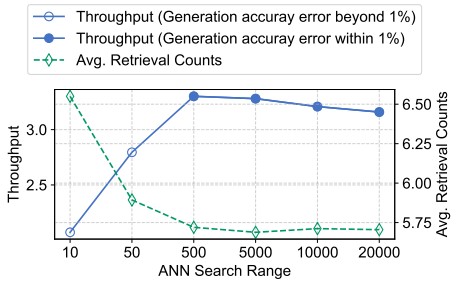

Figure 1: **Impact of Retrieval Accuracy on Search Agent Efficiency.** Higher ANN search range means higher-recall retrieval. Throughput marks the number of requests completed per second (higher is better). Retrieval count indicates the number of retrievals called per request. End-to-end generation accuracy error is calculated by comparison with an exact retrieval method.

However, further increasing the search range (e.g., beyond 10,000) yields diminishing returns. While average retrieval counts decrease slightly, throughput declines due to the higher cost of very high-recall ANN searches. This suggests that simply maximizing retrieval recall is not the optimal strategy for search agent efficiency. Once retrieval quality sufficiently supports reasoning, additional search efforts offer marginal benefits and can even harm overall efficiency.

### 2.2.2 IMPACT OF RETRIEVAL LATENCY

**Insight 2:** *Compared to naive RAG, search agents are much more sensitive to retrieval latency due to ignoring inter-request priorities and retrieval-induced stalls.*

For naive RAG, all requests are retrieved before generation. Retrieval latency (millisecond level) is negligible compared to the total request latency (second level), so it is insensitive to retrieval latency. However, for search agents, retrieval occurs during self-reasoning, where the time scale of a single token generation and retrieval latency are comparable. Minor retrieval latency can cause requests to be inserted into different token generation iterations, leading to different system behaviors.

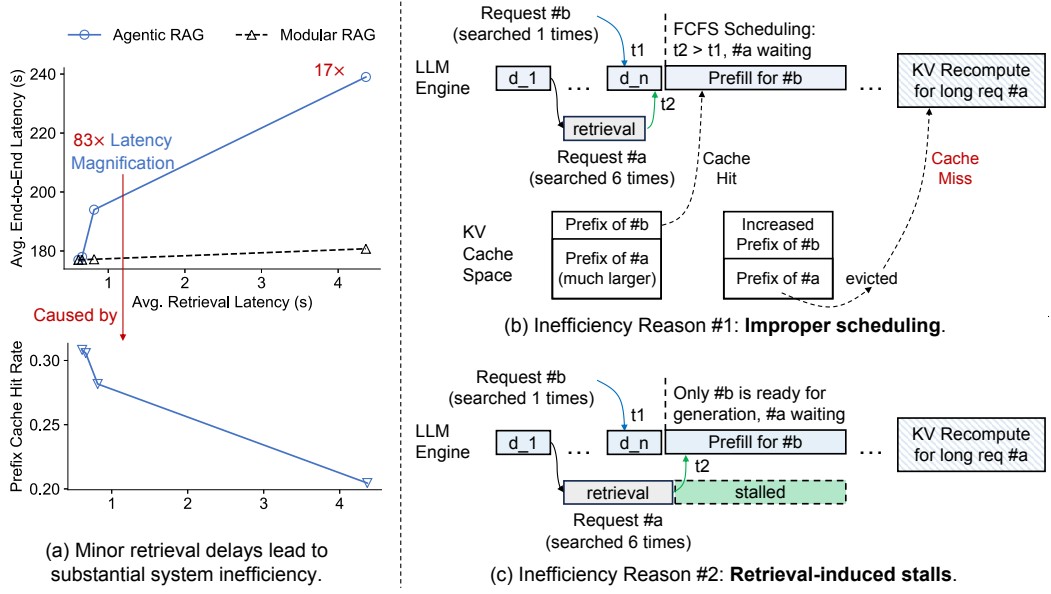

Figure 2: **Impact of Retrieval Latency on Search Agent Efficiency.** **(a)** Search agents exhibit significantly higher retrieval latency sensitivity than naive RAG (up to $83\times$ magnification), linked to lower prefix KV-cache hit rates. **(b, c)** Root causes include: **(b)** improper scheduling, where serving shorter requests first evicts cache for longer ones, causing recomputation; and **(c)** retrieval-induced stalls, where requests missing a scheduling point must wait, risking cache eviction.

Figure 2 shows the average end-to-end latency of search agents and RAG under different retrieval latency (controlled by different search ranges), with a request rate of 5 requests/second and a test duration of 10 minutes. For fair comparison, we assume RAG generates the same length of tokens with search agent, and its end-to-end latency $t^{e2e}$ is calculated as $t_0^{e2e} + \bar{t}_{ret} \cdot \bar{n}_{ret}$, with $t_0^{e2e}$ as the token generation time without retrieval, $\bar{t}_{ret}$ as the average retrieval time, and $\bar{n}_{ret}$ as the average retrieval counts per request. The results indicate that search agents suffer from drastic efficiency degradation under even minor retrieval delays. As average retrieval latency increases from $0.6$s to $4.4$s, the end-to-end latency of the search agent is magnified by over $83\times$, while RAG remains largely stable. This severe magnification in search agent is strongly correlated with a sharp decrease in the prefix KV-cache hit rate, dropping from over $30\%$ to under $21\%$, which forces frequent and costly KV recomputations (Figure 2a).

We identify two root causes for this observed behavior, both contributing to unnecessary KV recomputation, particularly for longer, multi-turn requests: **improper scheduling** and **retrieval-induced stalls**. Figure 2b illustrates the issue of improper scheduling. Consider request #a, which involves a longer reasoning path with 6 retrievals, and request #b, which just completes a single retrieval. Even if request #a arrives first, if its retrieval completes slightly later than that of #b ($t_2 > t_1$), a standard FCFS scheduler may choose to serve #b first in the next iteration. As #b proceeds with its generation, it occupies valuable KV-cache space, potentially leading to the eviction of the prefix KV-cache belonging to #a. When request #a eventually resumes, it encounters a *cache miss* and must recompute its entire prefix from scratch, significantly increasing its latency. Our measurements highlight the high cost of such improper scheduling: 55.9% of tokens were unnecessarily recomputed in affected cases, leading to more than a 108% increase in computation time per request.

Even with improved scheduling, another significant inefficiency risk comes from reasoning stalls, depicted in Figure 2c. The asynchronous execution of retrieval and generation can lead to subtle timing misalignments. If a long request like #a completes its retrieval only slightly after the deadline for inclusion in the next generation step, it misses the current scheduling batch and is forced to wait until the subsequent one. We term this unproductive waiting period "retrieval-induced stalls." During this stall, shorter requests (e.g., #b) that are ready can continue executing. Their execution may further displace #a's prefix from the KV-cache, once again resulting in costly recomputation upon

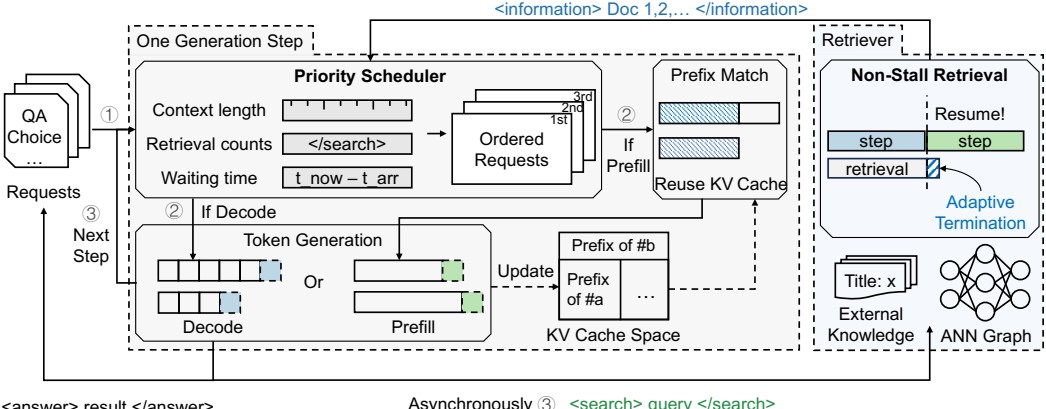

Figure 3: **SearchAgent-X's Architecture.** Requests are scheduled with priorities. Reasoning and retrieval are interleaved, with a non-stall retrieval mechanism to avoid unnecessary waiting.

#a's eventual resumption. Our data shows that, on average, more than 25% of sequences experience such stalls after completing their retrieval across various scenarios.

**Limitations of Existing Solutions.** Our analysis highlights key limitations in current search agent systems. ENN retrieval, despite full recall, incurs prohibitive retrieval overhead. While high-recall ANN search is more suitable, it suffers from retrieval-induced stalls due to asynchronous execution. Furthermore, prevalent FCFS scheduling in LLM inference frameworks (Kwon et al., 2023; Abhyankar et al., 2024) disregards the search agent's unique request priorities, leading to suboptimal cache utilization and costly recomputation.

## 3 DESIGN OF SEARCHAGENT-X

### 3.1 OVERALL ARCHITECTURE

Drawing upon the above insights, we propose SearchAgent-X, a simple yet efficient inference system that is easy to deploy, explicitly designed to optimize end-to-end efficiency for search agent workloads by smoothly interleaving self-reasoning and retrieval. Figure 3 shows SearchAgent-X's architecture, a tightly integrated system processing search agent requests at the token generation level. At each LLM output step, the system checks for special tags that trigger the Retriever for an ANN-based search (e.g., <search>) or request completion (e.g., <answer>), respectively.

To optimize GPU resource usage, SearchAgent-X incorporates a **priority scheduler**. It dynamically prioritizes concurrent requests using real-time collected metrics like retrieval count and waiting time, aiming to enhance KV-cache reuse by processing higher-priority requests first. During prefill, prefix matching reuses existing KV pairs from cache, significantly reducing computational overhead; new KV states are computed if caching is inapplicable or a miss occurs. Retrieval and generation operate asynchronously to enhance throughput. When retrieval is triggered, the system queries a pre-built ANN graph index. To proactively avoid retrieval-induced stalls, SearchAgent-X employs **non-stall retrieval** with adaptive search termination, allowing generation to proceed without unnecessary waiting while ensuring sufficient retrieval quality.

### 3.2 PRIORITY SCHEDULING

SearchAgent-X employs a priority-based scheduling mechanism to efficiently and fairly manage concurrent generation requests. As introduced earlier, each search agent request $i$ involves a list of generation sequences $[s_{i,0}, s_{i,1}, \ldots, s_{i,r_i}]$, where $s_{i,0}$ is the initial sequence and $s_{i,j}$ ($j > 0$) represents a sequence resumed after the $j$-th retrieval. Let $r_i$ denote the current number of retrievals performed for request $i$, and $s_{i,r_i}$ be the sequence currently being processed.

As discussed earlier, requests that have undergone more retrieval steps (i.e., higher $r_i$) benefit more significantly from prefix cache reuse due to longer shared prefixes. Prioritizing such requests can

therefore enhance overall cache efficiency and reduce redundant computation. However, scheduling solely based on retrieval count risks starving requests with fewer or no retrievals, leading to increased end-to-end latency and reduced fairness.

To mitigate these issues, `SearchAgent-X` utilizes a hierarchical scheduler that dynamically prioritizes requests based on a combination of three key metrics associated with request $i$: (1) the number of retrievals completed $R_i = r_i$; (2) the context length of the current sequence $C_i = L_{\text{seq},i}$; and (3) the waiting time of the initial request $W_i = t_{\text{now}} - t_{\text{arr},i}$. The first two metrics implicitly prioritize sequences with longer reusable prefixes, while the last ensures fairness by giving preference to requests that have been waiting longer overall.

Instead of combining these diverse metrics into a single weighted score, which would require tedious and potentially task-specific tuning of weights, `SearchAgent-X` discretizes each metric into $G$ distinct priority levels. For a given metric $M \in \{R, W, C\}$, the threshold defining the lower bound for level $k$ is calculated as:

$$T_{M,k} = \min(M) + \frac{k}{G} \cdot (\max(M) - \min(M)), \quad 0 \le k < G \tag{1}$$

A request $i$ is then assigned to the highest priority level $k$ for which at least one of its metric values $(R_i, W_i, C_i)$ exceeds the corresponding threshold $T_{M,k}$:

$$k = \max \left\{ j \in [0, G-1] \mid R_i > T_{R,j} \vee W_i > T_{W,j} \vee C_i > T_{C,j} \right\} \tag{2}$$

Requests that do not meet any threshold are assigned to the base level 0.

Within each assigned priority level, active sequences are further sorted according to their current queueing time, defined as $W_i^{\text{cur}} = t_{\text{now}} - t_{\text{arr},i}^{(r_i)}$, where $t_{\text{arr},i}^{(r_i)}$ is the time when the sequence $s_{i,r_i}$ becomes ready for processing (e.g., after retrieval completes). Sorting by $W_i^{\text{cur}}$ in descending order ensures that among requests of similar priority level, those that have been waiting longest for their current step are processed first, mitigating the risk of KV-cache eviction during extended waits.

Finally, `SearchAgent-X` determines the execution order by traversing the priority levels from highest to lowest and processing the sequences within each level based on their sorted $W_i^{\text{cur}}$.

## 3.3 Non-Stall Retrieval

To mitigate inefficiencies from retrieval latency and prevent retrieval-induced stalls (Section 2.2.2), `SearchAgent-X` incorporates a flexible, non-stall early termination strategy for Approximate Nearest Neighbor (ANN) search. Unlike traditional ANN search that iteratively refines candidates until meeting pre-set criteria (e.g., explored nodes, list stability) and can thus cause pipeline stalls if retrieval is slow, `SearchAgent-X` adaptively concludes the search. This adaptive termination is based on two key conditions: the maturity of retrieval results and the readiness of the LLM engine, ensuring generation proceeds without unnecessary blocking.

At the core of this strategy is the concept of a *soft limit* for the retrieval process. This soft limit represents a checkpoint where search results are likely to have achieved sufficient quality for the generation task. `SearchAgent-X` estimates retrieval maturity by monitoring returns in quality improvement during the ANN search. While retrieval quality generally improves with more explored neighbors, we find that the rate of improvement diminishes significantly after a certain point, exhibiting a "knee" where newly found points contribute less to the overall quality. `SearchAgent-X` exploits this observation. A normalized metric $\text{RQ}_t$ is used to evaluate the quality of newly discovered candidates at step $t$, defined as: $(d_t - d_{\text{best}})/(d_{\text{worst}} - d_{\text{best}})$, where $d_t$ is the new candidate's distance to the query, while $d_{\text{best}}$ and $d_{\text{worst}}$ are the distances of the best and worst candidates currently in ANN algorithm's list. A high $\text{RQ}_t$ value suggests the new candidate offers little improvement over existing ones, indicating diminishing returns from further search (see details in Appendix B.2).

The maturity exit criterion is met when this smoothed quality signal (derived from $\text{RQ}_t$) indicates a plateau (i.e., exceeds a threshold $\tau$) *and* the LLM engine is ready for its next token generation operation. Upon meeting both conditions, `SearchAgent-X` halts the retrieval and provides the current, sufficiently mature results to the LLM; otherwise, retrievals stop naturally. This adaptive alignment of asynchronous retrieval and generation significantly reduces end-to-end latency without compromising the quality of the retrieved context, contrasting with traditional fixed stopping criteria that may not optimally synchronize with the dynamic state of the generation pipeline. `SearchAgent-X`'s complete execution process and implementation details can be found in Appendix B.

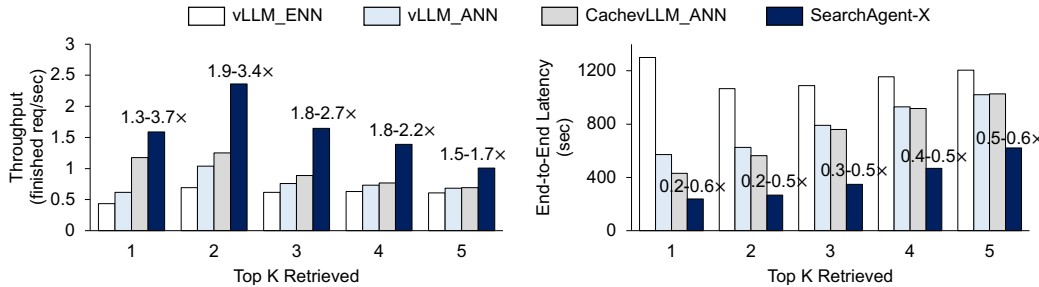

Figure 4: **End-to-End Efficiency of Offline Inference.** Left: Requests completed per second (higher is better). Right: Average end-to-end latency (lower is better). `SearchAgent-X` is our proposed system, which consistently outperforms all baselines across different top-$k$ values.

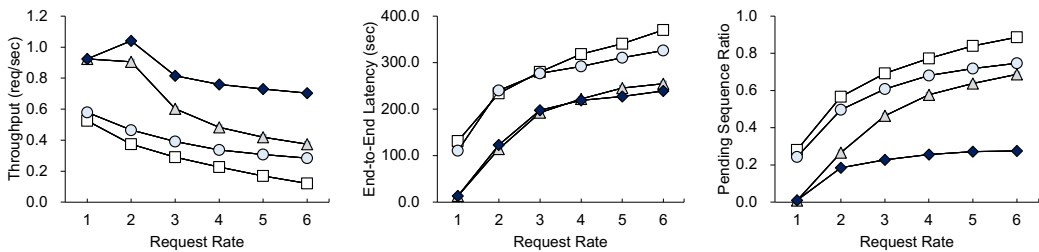

Figure 5: **End-to-End Efficiency of Online Inference.** Left: Throughput. Middle: Latency. Right: Pending Sequence Ratio, the percentage of sequences initiated but not completed within the test period. Lower is better, indicating reasonable workload scheduling.

# 4 EVALUATION

## 4.1 EXPERIMENTAL SETUP

**Models and Datasets.** We evaluate our method on two different search agent models from Search-R1 (Jin et al., 2025a) and ReCall (Chen et al., 2025). For retrieval, we adopt a chunked Wikipedia dataset as the knowledge base, using an ANN index constructed with HNSWlib (Malkov & Yashunin). Note that our approach is model-agnostic and readily generalizes to other reasoning models/ANN methods.

**Testbed.** For the 7B model, we use a single NVIDIA L20 GPU with 48GB memory. For the 14B model, we use two A100 GPUs with 40GB memory each, connected via PCIe 3.0. The retrieval system runs on 22 CPU cores with 120GB of RAM.

**Baselines.** We compare the performance of four methods: 1) `vLLM_ENN`: the vanilla vLLM with exact retrieval. 2) `vLLM_ANN`: vanilla vLLM system (Kwon et al., 2023) with approximate retrieval. 3) `CachevLLM_ANN`: vanilla vLLM with approximate retrieval and prefix cache. 4) `SearchAgent-X`: our proposed system. Refer to Appendix B.3 for detailed setup.

## 4.2 END-TO-END PERFORMANCE

We first evaluate the end-to-end performance of different methods. For efficiency measurement, we use Musique (Jin et al., 2025b) and HotpotQA (Yang et al., 2018), two datasets of complex multi-hop queries. Two scenarios are tested: (1) offline inference, where all requests arrive at the start; and (2) online inference, where requests arrive at a fixed rate. In the offline setting, we process 1000 requests and measure efficiency upon completion. In the online setting, requests arrive at rates from 1 to 6 over a 10-minute window. Results for the 7B Search-R1 model are shown in Figures 4 and 5; full results across all metrics and models are in Appendix C.1 and C.2.

**In offline scenarios, `SearchAgent-X` consistently outperforms all baselines in terms of system throughput and per-request latency.** As shown in Figure 4, `SearchAgent-X` achieves 1.3-3.4× higher throughput and only 0.2-0.6× the latency compared to the baselines across different top-$k$

Table 1: Generation Quality of `SearchAgent-X` and Exact Retrieval.

| Dataset | Musique | NQ | 2Wiki | HotpotQA | Bamboogle | StrategyQA | Avg. |
|---|---|---|---|---|---|---|---|
| **Generation Accuracy** | | | | | | | |
| Exact Retrieval | 0.203 | 0.316 | 0.371 | 0.429 | 0.472 | 0.788 | 0.430 |
| `SearchAgent-X` | 0.203 | 0.320 | 0.370 | 0.428 | 0.472 | 0.789 | 0.430 |

values. Even in the most challenging case of top-$k$=5, `SearchAgent-X` still beats the best baseline `CachevLLM_ANN`, with a significant margin (1.5× in throughput and 0.6× in latency). We attribute this improvement to the `SearchAgent-X`'s high-recall ANN, and the design of efficient scheduling and non-stall retrieval mechanisms. `vLLM_ENN` performs poorly in this scenario, as it incurs excessive retrieval overhead and hinders end-to-end reasoning efficiency. `vLLM_ANN` employs a high-recall ANN and performs obviously better than `vLLM_ENN`, but it still suffers from the inefficiencies of large amounts of recomputation due to the lack of prefix cache. `CachevLLM_ANN` uses prefix cache to reduce recomputation, but it still fails to wisely manage the scheduling of requests and avoid retrieval-induced stalls, leading to a significant performance gap compared to `SearchAgent-X`.

We also find that the performance of all methods first increases then decreases with the increase of top-$k$ values. This aligns well with our previous observation that both overly high and overly low retrieval quality degrade end-to-end efficiency. When the top-$k$ value is too small, the model may fail to retrieve useful documents, leading to longer reasoning sequences and lower throughput. Conversely, when the top-$k$ value is too large, the concatenated sequence becomes too long, resulting in longer prefill time. However, we note that `SearchAgent-X` consistently outperforms all baselines across all top-$k$ values, indicating its robustness to different retrieval settings.

We also observe that SearchAgent-X often yields larger gains on harder tasks. As shown in Table 3 in Appendix C.1, the HotpotQA dataset (a easier dataset according to the Search-R1 analysis) yields lower benefits compared to the Musique dataset in our main experiments (e.g., 1.52x of throughput improvement on easy dataset vs. 1.84x of throughput improvement on hard dataset, for top-k=3).

**In online scenarios, `SearchAgent-X` utilizes computing resources more efficiently than baselines, completing more requests in the same amount of time.** As shown in Figure 5, `SearchAgent-X` completes at least 1.5×, and up to 3.5× more requests on average than the baseline, within the request rate range of 1 to 6. Further, we record the pending sequence ratio that measures the resource utility of the system, defined as the percentage of sequences that are initiated but not completed within the test period. As shown in Figure 5 (right), `SearchAgent-X` achieves stable, small pending sequence ratios (about 0.2), while the baselines experiences dramatic increases with higher request rates (more than 0.6), indicating the effectiveness of `SearchAgent-X`'s scheduling.

**`SearchAgent-X` achieves comparable generation quality to exact retrieval.** We evaluate the generation quality of `SearchAgent-X` and exact retrieval (`vLLM_ENN`) on six representative datasets. We use the *Exact Match* metric as generation accuracy to measure the correctness of the generated answers (Jin et al., 2025b). As shown in Table 1, `SearchAgent-X` achieves similar generation accuracy, retrieval counts and output length (see Appendix C.2 for details) as exact retrieval across all datasets, indicating that it does not compromise generation quality for efficiency. Another interesting finding is that `SearchAgent-X` may even achieve higher generation accuracy on some datasets, such as NQ (0.320 vs. 0.316). The results could be attributed to two aspects. First, full ANN recall does not necessarily mean optimal generation accuracy; the correct answer document may not always be captured by semantic similarity. Second, the search agent model has the adaptability to adjust the reasoning length. ANN might lead search agents to perform an extra reasoning step (e.g., 2.292 vs. 2.288 for NQ), adjusting the retrieval query, which in turn improves generation accuracy.

## 4.3 ABLATION STUDY

**The priority scheduling and non-stall retrieval of `SearchAgent-X` help improve the prefix cache utility, thus enhancing end-to-end efficiency.** In this section, we use Search-R1 7B model on Musique dataset. Figure 6 (left) shows the end-to-end performance of different techniques for offline inference with top-$k$ = 5. We have several observations. First, the advantages of prefix cache are diminished in this challenging scenario. With top-$k$=5, it's only 1.01× that of `vLLM_ANN`, compared to

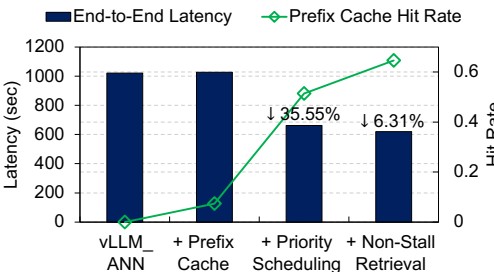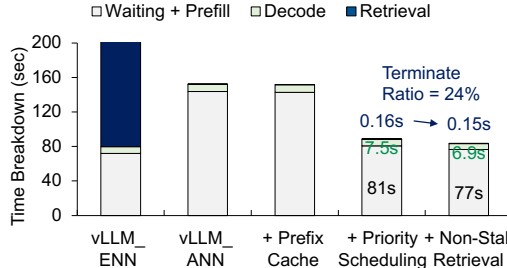

Figure 6: **SearchAgent-X Technique Breakdown for End-to-End Performance (Left) and Per-Sequence Generation Time (Right).** Each bar/scatter adds one technique over its left bar/scatter, with the leftmost being vanilla vLLM and the rightmost being the full SearchAgent-X.

$1.91\times$ with top-$k$=1. This validates that the benefits of prefix cache still require appropriate scheduling and retrieval methods to unleash its potential. Second, SearchAgent-X's priority scheduling reduces the end-to-end latency by 35.55% based on prefix cache. This is because the priority of requests is properly managed, maximizing the utilization of GPU resources. In addition, the prefix cache hit rate increases from 0.07 to 0.51, verifying the effectiveness of the technique. Third, SearchAgent-X's non-stall retrieval further improves the hit rate to 0.65, leading to a further 6.3% reduction in latency. This shows that the adaptive termination strategy fully utilizes the "free lunch" of asynchronous execution, timely recalling mature retrieval results, thereby improving system processing efficiency.

Figure 6 (right) further demonstrates the per-sequence generation time of different parts. We have more observations. First, for vLLM_ENN, the retrieval time is the largest component, accounting for over 60% of the total time. Instead, its prefill time is the lowest across different techniques, since its reasoning requires waiting for long-time retrieval, thus reducing the pressure on token generation. Second, for priority scheduling, we note that it reduces not only the prefill time (due to more prefix cache utilized), but also the decode time, showing a better system processing capability. This is because by improving KV-cache utilization, it avoids recomputation of long requests, freeing up GPU space earlier for better decode parallelism. Third, non-stall retrieval actually only reduces 0.01s of retrieval time (from 0.16s to 0.15s), with about 24% of the retrievals being early terminated, but significantly reduces the end-to-end latency (41s, the end-to-end latency shown in Figure 6 (left)). This aligns well with our previous observation that minor retrieval latency can cause drastic efficiency degradation (as shown in Figure 2). Non-stall retrieval adaptively terminates only a small set of retrievals when necessary, yet yields the significant benefit of better cache utilization. More experiments can be found in Appendix C.3 to C.7.

# 5 RELATED WORK

Several recent systems optimize one- or multi-round RAG pipelines, such as TELERAG (Lin et al., 2025), RAGO (Jiang et al., 2025), PATCHWORK (Hu et al., 2025), RAGServe (Ray et al., 2024), RAGCache (Jin et al., 2024), AquaPipe (Yu et al., 2025), AutoRAG (Fu et al., 2024), and PipeRAG (Jiang et al., 2024). These methods either prefetch or pipeline retrieval results, or tune retrieval/generation hyperparameters, but they still treat retrieval and generation as largely separated stages and do not analyze the root causes of inefficiency in the dynamically interleaved search–reasoning pattern targeted by SearchAgent-X.

Techniques that could change sequence prefix, such as context trimming, summarization, or token sparsification (Jiang et al., 2023) are also compatible with our design: as long as they are applied in a prefix-preserving fashion during decoding, prefix caching remains effective and our scheduler can simply use the effective cached prefix length (after sparsification) as a scheduling feature.

Meanwhile, broader agent workflow optimizations, such as auto-tuning (He et al., 2025), KV-cache management (Abhyankar et al., 2024), and partial tool execution (Xu et al., 2024), improve overall efficiency but overlook the specific challenges of retrieval accuracy and latency in search agents. In contrast, SearchAgent-X directly addresses these challenges by tightly coupling priority-scheduled reasoning with non-stall retrieval, yielding improved efficiency. Notably, our approach is orthogonal

to, and can potentially be combined with, other RAG optimization techniques such as context compression (Shi et al., 2024) and retrieval reranking (Glass et al., 2022).

## 6    CONCLUSION

LLM reasoning-driven search agents offer great potential for complex problems, but face severe, distinct efficiency challenges. This paper highlights the non-trivial impact of retrieval accuracy and the latency sensitivity caused by scheduling deficiencies and retrieval-induced stalls. Our proposed system, `SearchAgent-X`, designed based on these insights, demonstrates substantial improvement in system efficiency, all while maintaining high generation quality. This study provides important insights for practical deployments of high-efficiency LLM-based search agents, and the proposed solutions are easily adaptable to other ANN retrieval methods and LLM reasoning models.

## REPRODUCIBILITY STATEMENT

We provide the complete source code of SearchAgent-X in an anonymous repository at `https://github.com/tiannuo-yang/SearchAgent-X`. The repository is self-contained with instructions and scripts to replicate all experiments, along with dataset specifications and pre-trained model references.

## ACKNOWLEDGMENTS

W.N. is supported by the National Science Foundation (NSF) under Award No. 2427856.

T.Y. thanks his friend Ke Cheng for the technical advice and discussion provided during the rebuttal.

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

# APPENDIX

## A  AN ILLUSTRATION OF LLM-BASED SEARCH AGENTS

Figure 7 shows an example of a search agent process. Faced with a complex query ("Curious is a women's fragrance by a singer born in what city and state?"), the search agent first engages in preliminary reasoning ("I need to find out which city and state a singer..."). Recognizing a knowledge gap regarding the "Curious fragrance," the model proactively decides to initiate a search ("search Curious fragrance information"). Upon receiving the crucial information ("Curious is a women's fragrance by Britney Spears"), the model doesn't conclude its process. Instead, it integrates this new knowledge into its subsequent thought process and reasoning. This triggers further searches, of which the retrieval result is concatenated with previously generated tokens and re-injected into LLMs. Through this dynamic cycle of "think-search-rethink," the model progressively assembles the necessary pieces of the knowledge puzzle required to answer the question fully. This culminates in a high-quality answer that addresses all aspects of the initial query ("McComb, Mississippi"). This ability to autonomously plan retrieval actions and iteratively incorporate new information into its reasoning process allows the search agent to tackle more complex questions and deliver better responses, moving beyond reliance solely on pre-trained knowledge or a single retrieval.

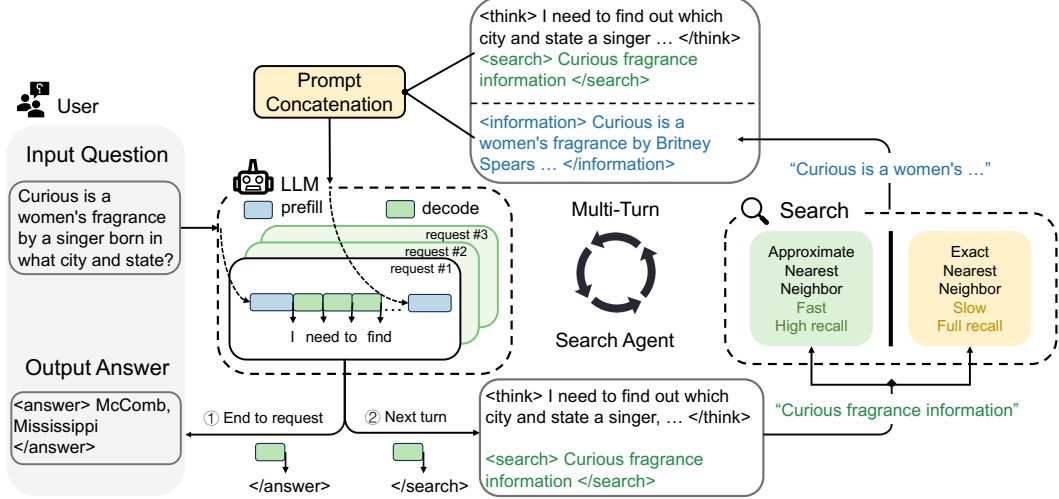

Figure 7: An illustration of reasoning and search interleaved LLM-based search agents.

## B  IMPLEMENTATION DETAILS

### B.1  SEARCHAGENT-X EXECUTION

This section outlines the high-level execution flow of the `SearchAgent-X` system, as depicted in Algorithm 1, complementing the conceptual component descriptions in Section 3 of the main paper. `SearchAgent-X` orchestrates LLM inference (with prefix caching, Section 3.1), dynamic high-recall approximate retrieval, Priority Scheduling (Section 3.2), and Non-Stall Retrieval (Section 3.3) to achieve efficient search agents. The system initializes an LLM inference engine and manages incoming requests, active asynchronous search tasks, and their results.

The main execution loop begins by ingesting new user requests into the LLM engine's pool (line5-7). Concurrently, if Non-Stall Retrieval is active, `SearchAgent-X` consults an external signal to identify and expedite the completion of any ongoing retrieval tasks that have reached sufficient maturity or for which LLM engine readiness dictates early termination (line 10-11), thus preventing pipeline stalls. Upon completion of a search (either normally or via early termination), retrieved documents are concatenated with the original context, and the augmented request is resubmitted to the LLM engine (line 15-20).

The core of the processing loop involves LLM generation and the agentic control flow. Before each LLM generation step, `SearchAgent-X`'s Priority Scheduling policy is applied to the queue of waiting requests, reordering them to optimize system throughput and KV-cache utilization (line 23). Following token generation by the LLM, the output for each active sequence is parsed (line 25-28). If a `<search>` tag is detected, indicating a need for external knowledge, `SearchAgent-X` halts further generation for that sequence and launches an asynchronous high-recall ANN retrieval task (line 29-34). Conversely, if a `<answer>` tag is identified or the sequence naturally concludes, the request is finalized (line 35-38). This iterative and asynchronous process enables the dynamic interleaving of LLM reasoning, external knowledge retrieval, and intelligent scheduling, which is fundamental to `SearchAgent-X`'s efficient handling of complex search agent workloads.

---

**Algorithm 1** `SearchAgent-X` Main Execution Loop

---

1: Initialize LLM_Engine, ArrivalQueue, ActiveSearchTasks, FinishedOutputs
2: Configure PriorityScheduling (enabled/type), NonStallRetrieval (enabled)
3: **while** LLM_Engine has unfinished requests **or not** ActiveSearchTasks is empty **or not** ArrivalQueue is empty **do**
4:     // Step 1: Ingest new requests
5:     **for** each request $R_{new}$ in ArrivalQueue ready for processing **do**
6:         Add $R_{new}$ to LLM_Engine's request pool
7:         Remove $R_{new}$ from ArrivalQueue
8:     **end for**
9:     // Step 2: Non-Stall Retrieval Check (if enabled)
10:     **if** NonStallRetrieval is enabled **and** ActiveSearchTasks is not empty **and** LLM_Engine has waiting requests **then**
11:         TerminatedSearchIDs ← CheckExternalNonStallSignal()
12:         // Queries for searches to terminate early
13:     **end if**
14:     // Step 3: Process completed search tasks
15:     **for** each search task $S_i$ in ActiveSearchTasks that has completed **do**
16:         $R_{orig}$, retrieved_docs, search_finish_time ← $S_i$.getResult()
17:         new_context ← Concatenate($R_{orig}$.context, retrieved_docs)
18:         AddResumedRequest($R_{orig}$, new_context, search_finish_time) to LLM_Engine
19:         Remove $S_i$ from ActiveSearchTasks
20:     **end for**
21:     // Step 4: LLM Generation Step
22:     **if** LLM_Engine has unfinished requests **then**
23:         // Section 3.2
24:         ApplyPriorityScheduling(LLM_Engine.waiting_requests)
25:         LLM_Outputs, Scheduler_Info ← LLM_Engine.step()
26:         RecordTokenTimingsAndPrefixCacheStats(Scheduler_Info)
27:         **for** each output $O_j$ in LLM_Outputs **do**
28:             current_text ← $O_j$.getGeneratedText()
29:             **if** DetectSearchTag(current_text) **then**
30:                 query ← ExtractSearchQuery(current_text)
31:                 LLM_Engine.abortRequest($O_j$.request_id)
32:                 $S_{new}$ ← LaunchAsyncRetrievalTask($O_j$.request_id, current_text, query)
33:                 // High-recall ANN
34:                 Add $S_{new}$ to ActiveSearchTasks
35:             **else if** DetectAnswerTag(current_text) **or** $O_j$.isFinished() **then**
36:                 Add $O_j$ to FinishedOutputs
37:                 LLM_Engine.abortRequest($O_j$.request_id)
38:             **end if**
39:         **end for**
40:     **end if**
41: **end while**
42: CleanupRemainingTasks()

---

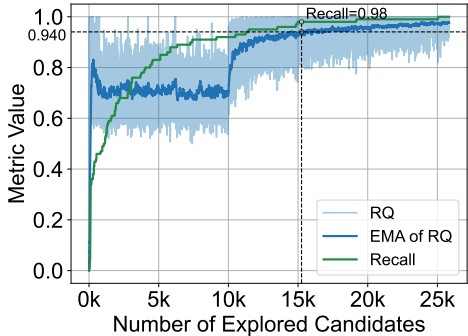 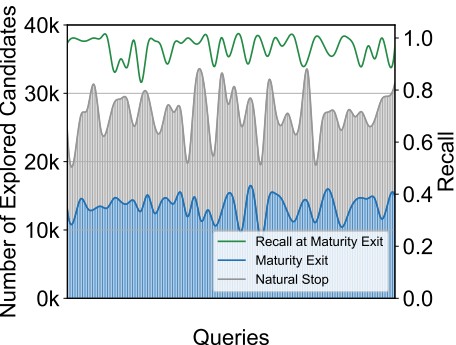

Figure 8: **EMA Signal of Retrieval Maturity.** RQ means relative quality of newly explored candidates. EMA represents smoothed RQ. The vertical line marks maturity, where the improvement of recall and EMA decreases to about 0.

Figure 9: **Comparison of Maturity Exit and Natural Stop.** The shadows represent the number of candidates explored by two methods (showing a similar trend). The curve represents the recall of maturity exit (consistently high).

## B.2 RETRIEVAL MATURITY ESTIMATION

In practice, the raw $RQ_t$ signal described in Section 3.3 exhibits short-term fluctuations that may hinder robust maturity estimation. To address this, `SearchAgent-X` applies an exponential moving average (EMA) (Holt, 2004), with a window size of 500, to smooth the signal.

Selecting an appropriate threshold $\tau$ for the EMA is critical to balancing retrieval quality and latency. To determine a suitable value, we sample queries from the Musique dataset and record the evolution of the EMA curve as the number of explored candidates increases. For each query, we identify the point where the EMA curve flattens—i.e., where marginal improvements approach zero—indicating that newly explored candidates are far from the query and contribute little to quality. This point reflects the onset of retrieval maturity. As shown in Figure 8, the recall at this stage also stabilizes and reaches a high level (around 0.98). We adopt the corresponding EMA value ($\tau = 0.9$) at this "knee" as the practical threshold $\tau$ in `SearchAgent-X` to reliably trigger maturity exit. We provide more analysis of this design to verify its effectiveness in Appendix C.3.

## B.3 DETAILED EXPERIMENTAL SETUP

We implement `SearchAgent-X` by building upon vLLM (Kwon et al., 2023), a state-of-the-art LLM inference engine to use its efficient PagedAttention mechanism. For retrieval component, we use a knowledge base constructed from a chunked Wikipedia dataset, containing approximately 21 million text chunks. Each chunk is embedded into a 384-dimensional vector using the `all-MiniLM-L6-v2` model (Reimers & Gurevych, 2019). An Approximate Nearest Neighbor (ANN) index is built offline over these embeddings using HNSWlib (Malkov & Yashunin), configured with parameters such as up to 32 neighbors per node and an efConstruction (candidate list size during build) of 500. This index serves as the foundation for all ANN-based retrieval methods in our experiments. For these ANN searches (employed by `SearchAgent-X` and approximate retrieval baselines), we generally set the search range (e.g., efSearch) to 10,000 to achieve high recall with acceptable computational overhead, based on empirical analysis. These HNSW ANN searches leverage inter-query parallelism with 4 threads to optimize throughput while managing memory access contention.

Specific configurations for the different systems are as follows. For the exact retrieval baseline (`vLLM_ENN`), we adapt HNSWlib to perform exhaustive search more efficiently by enabling intra-query parallelism, utilizing 6 threads. For `SearchAgent-X`, beyond using the aforementioned high-recall ANN setup, we set its unique parameters: the priority scheduling level $G$ is configured to 6 (we note that `SearchAgent-X` exhibits low sensitivity to this specific value, as shown in our ablation study in Appendix C.4). The threshold $\tau_{EMA}$ for estimating retrieval maturity in the non-stall mechanism is set to 0.9, determined via offline profiling detailed in Appendix B.2. The approximate retrieval

baselines (`vLLM_ANN`, `CachevLLM_ANN`) also utilize the general ANN search settings described above, including the search range of 10,000 and 4 threads for inter-query parallelism.

## C  MORE RESULTS

### C.1  DETAILED OVERALL EFFICIENCY

We note that `SearchAgent-X` outperforms all baselines in different scenarios including model sizes, deployment methods (single GPU or distributed GPUs), top-$k$ values, and query datasets.

**Different Model Sizes and Top-k Values**   Table 2 presents detailed results of overall efficiency across different methods for Search-R1 under 7B (on a single L40 GPU) and 14B models (on two 40GB A100 GPUs). The advantage of `SearchAgent-X` is bigger in the 7B model and small top-$k$ values of the 14B model. This is due to two reasons. First, although the 14B model distributes model weights across two A100s, its KV-cache space is still smaller than the 7B model (because the model weights are larger, resulting in a larger KV-cache for a single token). The 7B model has a larger available KV-cache space, thus yielding greater benefits from managing the prefix cache. Second, the 14B model we test calls retrievals more cautiously, while the 7B model calls more retrievals (e.g., 4.9 of the 7B model vs. 3.3 of the 14B model when top-$k$ = 3 for Musique dataset), resulting in a greater distinction in request priority.

**Different Datasets**   Moreover, as shown in Table 3, on the HotpotQA dataset, SearchAgent-X also outperforms all baselines, achieving 1.47x to 2.55x higher throughput and 1.62x to 2.81x lower latency compared to the strongest baseline, `CachevLLM_ANN`.

**Different Search Agent Models**   To further demonstrate the generality of our method, we apply it to a different search agent architecture, ReCall (Chen et al., 2025). We observe that SearchAgent-X consistently delivers best efficiency. As shown in Table 4 below, when applied to ReCall's 7B model, `SearchAgent-X` attains 1.12x to 1.74x higher throughput, 1.31x to 1.81x lower latency, and 1.49x to 2.86x higher cache hit rate than baselines. The results demonstrate that `SearchAgent-X` still outperforms the most competitive baseline in this different search agent architecture, confirming the generality of SearchAgent-X.

### C.2  DETAILED GENERATION QUALITY

Table 5 further shows the generation details beyond Table 1 of `SearchAgent-X` and exact retrieval (`vLLM_ENN`) across different datasets. We note that `SearchAgent-X` achieves similar generation length as exact retrieval across all datasets (6822 tokens vs. 6826 tokens), indicating that our non-stall retrieval does not lead to unusual search agent model/retriever behaviours, making SearchAgent-X possible for maintaining perfect generation quality.

### C.3  ANALYSIS OF MATURITY EXIT MECHANISM

**The maturity exit mechanism effectively halts unnecessary searches without compromising retrieval quality.** To validate the effectiveness of non-stall retrieval, we analyze whether the maturity-based termination reliably halts unnecessary ANN iterations. We compare the retrieval traces of representative queries under two settings: maturity-based early stop and standard natural stop. As shown in Figure 9, we make several observations. First, query difficulty varies significantly across the dataset, resulting in different numbers of candidate nodes explored before natural convergence. This highlights the need for an adaptive termination strategy rather than relying on a fixed exploration budget. Second, for queries of varying difficulty, the number of candidates explored by the maturity-based strategy closely matches the natural termination point of standard search, indicating our maturity exit accurately captures query difficulties. More importantly, the recall achieved by maturity-stopped queries remains consistently high (0.963 on average). These results confirm that our non-stall retrieval effectively terminates redundant search iterations while preserving retrieval quality.

Table 2: Comparison across seven key metrics and top-$k$ values for different methods. Throughput and efficiency gains are marked by $\times$ multipliers. Lower values are better for metrics marked with $\downarrow$.

| Metric | Top-1 | Top-2 | Top-3 | Top-4 | Top-5 | Top-1 | Top-2 | Top-3 |
|---|---|---|---|---|---|---|---|---|
| | | | Search-R1-7B | | | | Search-R1-14B | |
| **Throughput** | | | | | | | | |
| vLLM_ENN | 0.44 | 0.69 | 0.62 | 0.63 | 0.61 | 0.46 | 0.47 | 0.43 |
| vLLM_ANN | 0.62 | 1.04 | 0.76 | 0.73 | 0.68 | 0.94 | 0.77 | 0.61 |
| CachevLLM_ANN | 1.18 | 1.25 | 0.89 | 0.77 | 0.69 | 1.08 | 0.89 | 0.70 |
| SearchAgent-X | 1.59 | 2.36 | 1.64 | 1.39 | 1.01 | 1.40 | 1.09 | 0.76 |
| Max Ratio | 3.61× | 3.42× | 2.65× | 2.20× | 1.66× | 3.04× | 2.32× | 1.77× |
| Min Ratio | 1.35× | 1.89× | 1.84× | 1.81× | 1.46× | 1.30× | 1.22× | 1.09× |
| **Token Throughput** | | | | | | | | |
| vLLM_ENN | 69.90 | 90.85 | 86.12 | 97.79 | 84.26 | 156.35 | 127.28 | 111.21 |
| vLLM_ANN | 101.21 | 136.81 | 106.09 | 114.28 | 94.65 | 320.76 | 206.46 | 159.70 |
| CachevLLM_ANN | 191.65 | 164.46 | 124.59 | 119.17 | 95.64 | 366.85 | 239.01 | 182.60 |
| SearchAgent-X | 259.94 | 309.73 | 229.96 | 216.21 | 139.25 | 472.76 | 292.13 | 199.14 |
| Max Ratio | 3.72× | 3.41× | 2.67× | 2.21× | 1.65× | 3.02× | 2.30× | 1.79× |
| Min Ratio | 1.36× | 1.88× | 1.85× | 1.81× | 1.46× | 1.29× | 1.22× | 1.09× |
| **Latency** $\downarrow$ | | | | | | | | |
| vLLM_ENN | 1300.56 | 1066.05 | 1089.37 | 1154.62 | 1205.16 | 1642.86 | 1567.85 | 1614.71 |
| vLLM_ANN | 571.36 | 625.33 | 790.68 | 930.29 | 1020.46 | 767.46 | 923.74 | 1052.74 |
| CachevLLM_ANN | 429.60 | 562.47 | 759.57 | 916.58 | 1026.91 | 673.35 | 805.60 | 980.35 |
| SearchAgent-X | 238.00 | 266.50 | 347.14 | 466.78 | 620.07 | 502.10 | 690.20 | 939.42 |
| Max Ratio | 0.18× | 0.25× | 0.32× | 0.40× | 0.51× | 0.31× | 0.44× | 0.58× |
| Min Ratio | 0.55× | 0.47× | 0.46× | 0.51× | 0.60× | 0.75× | 0.86× | 0.96× |
| **P99 Latency** $\downarrow$ | | | | | | | | |
| vLLM_ENN | 1758.64 | 1441.18 | 1462.45 | 1569.80 | 1641.90 | 2205.26 | 2136.75 | 2237.55 |
| vLLM_ANN | 915.04 | 956.46 | 1160.86 | 1348.87 | 1459.13 | 1066.63 | 1334.67 | 1555.47 |
| CachevLLM_ANN | 609.70 | 797.46 | 1073.03 | 1296.03 | 1446.88 | 930.06 | 1159.02 | 1454.61 |
| SearchAgent-X | 373.20 | 421.32 | 566.28 | 716.88 | 993.98 | 732.16 | 958.78 | 1362.24 |
| Max Ratio | 0.21× | 0.29× | 0.39× | 0.46× | 0.61× | 0.33× | 0.45× | 0.61× |
| Min Ratio | 0.61× | 0.53× | 0.53× | 0.55× | 0.69× | 0.79× | 0.83× | 0.94× |

Table 3: Comparison of system efficiency under the HotpotQA dataset.

| TopK | Method | Throughput | Latency | Token Throughput | Cache Hit Rate |
|---|---|---|---|---|---|
| 3 | vLLM_ENN | 0.78143 | 483.4094 | 106.0500 | 0.0000 |
| | CachevLLM_ANN | 1.30926 | 282.9210 | 179.9000 | 0.3890 |
| | SearchAgent-X | 1.99253 | 171.9478 | 271.8200 | 0.8890 |
| 4 | vLLM_ENN | 0.74530 | 505.7417 | 102.6500 | 0.0000 |
| | CachevLLM_ANN | 0.95950 | 375.9803 | 133.7800 | 0.3040 |
| | SearchAgent-X | 1.57188 | 210.8670 | 217.4600 | 0.9100 |
| 5 | vLLM_ENN | 0.68400 | 543.9000 | 87.5970 | 0.0000 |
| | CachevLLM_ANN | 0.81700 | 443.1300 | 105.2410 | 0.1706 |
| | SearchAgent-X | 1.20400 | 273.6900 | 154.8331 | 0.7340 |

Table 4: Comparison of system efficiency under the ReCall search agent model.

| TopK | Method | Throughput | Latency | Token Throughput | Cache Hit Rate |
|------|--------|-----------|---------|------------------|----------------|
| 3 | vLLM_ENN | 0.6370 | 376.375 | 224.550 | 0.000 |
| | CachevLLM_ANN | 0.9861 | 273.360 | 342.490 | 0.570 |
| | SearchAgent-X | 1.1035 | 207.920 | 391.898 | 0.849 |
| 4 | vLLM_ENN | 0.6200 | 423.070 | 205.630 | 0.000 |
| | CachevLLM_ANN | 0.8242 | 342.870 | 276.810 | 0.383 |
| | SearchAgent-X | 1.0780 | 234.390 | 364.066 | 0.804 |
| 5 | vLLM_ENN | 0.7180 | 393.390 | 220.810 | 0.000 |
| | CachevLLM_ANN | 0.7566 | 383.490 | 236.660 | 0.250 |
| | SearchAgent-X | 0.9570 | 282.085 | 302.049 | 0.716 |

Table 5: Generation Quality of `SearchAgent-X` and Exact Retrieval.

| Dataset | Musique | NQ | 2Wiki | HotpotQA | Bamboogle | StrategyQA | Avg. |
|---------|---------|-----|-------|----------|-----------|------------|------|
| **Retrieval Counts** | | | | | | | |
| Exact Retrieval | 3.247 | 2.288 | 3.126 | 2.702 | 2.440 | 2.496 | 2.717 |
| SearchAgent-X | 3.251 | 2.292 | 3.138 | 2.699 | 2.448 | 2.476 | 2.717 |
| **Output Length** | | | | | | | |
| Exact Retrieval | 8125 | 5839 | 7575 | 6840 | 6152 | 6402 | 6822 |
| SearchAgent-X | 8134 | 5847 | 7600 | 6839 | 6151 | 6382 | 6826 |

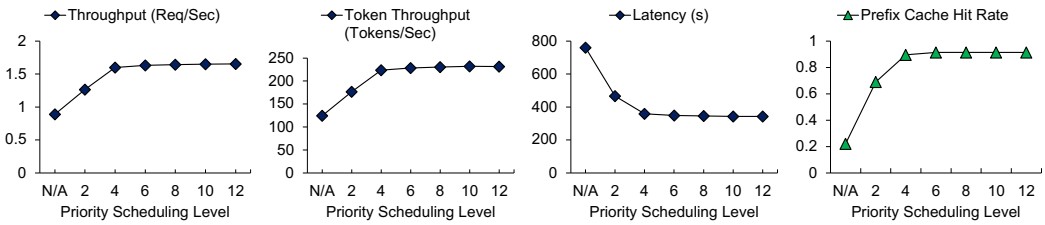

Figure 10: **Comparison of Different Priority Levels $G$ on Musique dataset.** The numbers on the X-axis represent different priority scheduling levels; N/A indicates that priority scheduling is not used.

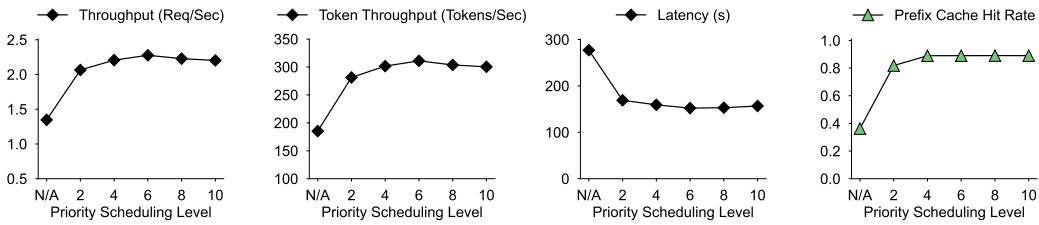

Figure 11: **Comparison of Different Priority Levels $G$ on HotpotQA dataset.** The numbers on the X-axis represent different priority scheduling levels; N/A indicates that priority scheduling is not used.

Table 6: System performance under different concurrency for vLLM and SearchAgent-X.

| Concurrency | 1 | 10 | 50 | 100 | 200 | 300 | 400 | 500 |
|---|---|---|---|---|---|---|---|---|
| **Cache Hit Rate** | | | | | | | | |
| vLLM_ANN | 0.131 | 0.065 | 0.088 | 0.135 | 0.168 | 0.173 | 0.173 | 0.177 |
| SearchAgent-X | 0.999 | 0.920 | 0.920 | 0.920 | 0.920 | 0.589 | 0.555 | 0.495 |
| **End-to-End Latency** | | | | | | | | |
| vLLM_ANN | 2181.75 | 633.37 | 477.88 | 448.64 | 437.12 | 425.89 | 415.44 | 411.67 |
| SearchAgent-X | 1378.82 | 323.21 | 212.05 | 199.29 | 208.07 | 295.39 | 314.80 | 327.87 |
| **Throughput** | | | | | | | | |
| vLLM_ANN | 0.190 | 0.604 | 0.782 | 0.843 | 0.853 | 0.865 | 0.862 | 0.865 |
| SearchAgent-X | 0.203 | 0.848 | 1.348 | 1.517 | 1.582 | 1.137 | 1.100 | 1.060 |

## C.4 COMPARISON OF DIFFERENT PRIORITY LEVELS $G$

**The performance of SearchAgent-X is insensitive to its priority level setting.** The priority level $G$ mentioned in Section 3.2 is an important hyperparameter of our method. In this section, we conduct an ablation study to evaluate the performance of SearchAgent-X with different priority levels $G$. The evaluation is performed on the Musique and HotpotQA datasets using the Search-R1 7B model in offline scenarios. We set $G = 2, 4, 6, 8, 10$, and 12, and compare them with the baseline without priority scheduling (N/A). The results are shown in Figure 10. We note that the performance of SearchAgent-X is not sensitive to the choice of $G$, and all efficiency metrics (including throughput, token throughput, latency, and prefix cache hit rate) first improve and then stabilize after $G = 6$. This is expected because the average retrieval number of the 7B model is within 4 and 6, while the primary objective of priority scheduling is to distinguish requests with different retrieval numbers for effective management.

We further validate this observation on the HotpotQA dataset. As shown in Figure 11, the efficiency consistently improves as $G$ increases (e.g., up to a $1.62\times$ throughput gain), and stabilizes once $G \geq 6$. This again confirms that the performance of SearchAgent-X is not sensitive to the choice of $G$, making it straightforward to tune across different settings.

## C.5 COMPARISON OF DIFFERNET CONCURRENCY SETTINGS

**SearchAgent-X consistently outperforms baselines across different concurrency levels.** To explore the impact of concurrency levels on system performance, we conduct an analysis to validate the effectiveness of SearchAgent-X under explicitly controlled request concurrency by tuning key parameters: iteration-level concurrency ($max\_num\_seq$) in vLLM (Kwon et al., 2023). The evaluation is conducted on the Musique dataset using the Search-R1 7B model in offline scenarios.

As detailed in Table 6, when increasing concurrency from 1 to 500, SearchAgent-X maintains a high cache hit rate up to around 200. Beyond that point, the cache hit rate drops more significantly. Meanwhile, system efficiency first increases then declines, showing a turning point: latency is minimized around concurrency = 100, and throughput peaks at concurrency = 200. This suggests that the GPU becomes saturated or experiences resource contention when concurrency exceeds 300. In practice, we typically choose the default value 256, which yields performance close to the optimal. The results demonstrate that SearchAgent-X consistently outperforms vLLM in both throughput and latency while exhibiting a tradeoff between cache hit rate and concurrency. We also observe that vLLM consistently suffers from low cache hit rates and lower system efficiency compared to SearchAgent-X across all concurrency levels. This is because suboptimal scheduling and retrieval under interleaved reasoning-retrieval workloads, where cache for previous tokens is frequently evicted—even with relatively small concurrency—causing inefficient reuse.

## C.6 COMPARISON WITH A NAIVE EARLY-STOPPING BASELINE

SearchAgent-X aligns stopping decisions with the LLM's prefill/decode progress, making the stop decision more timely and effective. In contrast, naive early-stopping approaches make halting

Table 7: Performance comparison of vanilla early stop and `SearchAgent-X`'s Non-Stall Retrieval.

| Method | Throughput (req/sec) | Latency (sec) | Cache Hit Rate | Accuracy |
|---|---|---|---|---|
| Vanilla HNSW | 0.7872 | 661.85 | 0.412 | 0.15 |
| Early Stop | 0.8050 | 634.77 | 0.564 | 0.15 |
| `SearchAgent-X` | 0.9502 | 620.07 | 0.645 | 0.15 |

Table 8: Comparison of system efficiency under the NSW search engine.

| TopK | Method | Throughput | Latency | Token Throughput | Cache Hit Rate |
|---|---|---|---|---|---|
| 3 | `vLLM_ENN` | 0.7880 | 395.040 | 97.640 | 0.000 |
| | `CachevLLM_ANN` | 1.7401 | 173.026 | 216.970 | 0.666 |
| | `SearchAgent-X` | 2.1270 | 129.862 | 265.690 | 0.891 |
| 4 | `vLLM_ENN` | 0.7196 | 578.940 | 99.487 | 0.000 |
| | `CachevLLM_ANN` | 1.0706 | 285.390 | 147.590 | 0.350 |
| | `SearchAgent-X` | 1.6560 | 172.340 | 227.730 | 0.864 |
| 5 | `vLLM_ENN` | 0.6940 | 428.642 | 87.531 | 0.000 |
| | `CachevLLM_ANN` | 0.8959 | 333.206 | 115.081 | 0.281 |
| | `SearchAgent-X` | 1.1402 | 241.234 | 146.360 | 0.690 |

decisions based exclusively on retrieval-side signals, such as a machine learning model that predicts the "maturity" of retrieval, while disregarding the state of LLM inference. To further validate the effectiveness of `SearchAgent-X`'s non-stall retrieval, we conduct a comparison against a vanilla early-stop baseline that only relies on retrieval status to decide the stopping point. The evaluation is conducted on the Musique dataset using the Search-R1 7B model with Top-5 retrieval in offline scenarios. Table 7 shows `SearchAgent-X` outperforms vanilla HNSW and the early-stop baseline. Early-stop baseline (Li et al., 2020) boosts throughput by 2.26% and cache hit rate by 0.152, cutting latency by 27.08 sec while keeping accuracy at 0.15. `SearchAgent-X` further increases throughput by 18.04% and cache hit rate by 0.081, reducing latency by 14.70 sec, with accuracy unchanged at 0.15. This confirms the efficiency gains of `SearchAgent-X`'s non-stall retrieval via aligned LLM scheduling and retrieval.

### C.7 SCALABILITY TO OTHER RETRIEVAL METHODS

To further validate the generalizability of our method, we implemented and evaluated it on an alternative retrieval backend, NSW (Malkov et al., 2014), which is another widely-used approximate search method. The experiments are conducted on the Musique dataset in offline scenarios using the Search-R1 7B model. As shown in Table 8, our method boosts the throughput of the NSW-based system by up to 2.7× and the cache hit rate by 2.46×, while reducing latency by 3.04×. These results confirm that our method is robust across retrieval strategies and our design is compatible with any iterative retriever, including cluster-based (e.g., IVF (Johnson et al., 2021)) and graph-based (e.g., HNSW (Malkov & Yashunin, 2018)) methods.

## D DISCUSSION

### D.1 APPLICABILITY

**Other Inference Engines** Our implementation of SearchAgent-X is a direct modification of vLLM. We therefore compare against vLLM-based baselines so that all systems share the same serving stack (PagedAttention, batching logic, etc.), avoiding interference with the performance observations from unrelated optimizations. However, The proposed optimization techniques are not specific to vLLM. SearchAgent-X only assumes: 1) token-level generation scheduling, and 2) an FCFS (or

FCFS-like) base policy that can be overridden. SGLang also adopts token-level generation scheduling and an FCFS-style policy, so SearchAgent-X can be integrated into SGLang with minimal changes for system improvement.

**Beyond Graph-Based ANN** For non-graph indices such as IVF or PQ-based methods, we treat probing additional clusters or codebook entries as units of search effort and apply the same maturity-based stopping rule; in large-scale industrial deployments it is common to tune these parameters to trade recall against latency (Wang et al., 2021; Ren et al., 2025), making SearchAgent-X readily applicable in such environments.

**Search APIs** While non-stall retrieval assumes controllable local searches, it is still possible to initiate concurrent requests to search APIs with multiple different search parameters, and do non-stall retrieval with an asynchronous manner as our compatible extension.

## D.2 FUTURE EXTENSIONS

**Multi-Agents** Our priority-aware scheduling and non-stall retrieval mechanisms offer a natural starting point for coordinating across agents, for example by: 1) introducing agent-level priorities or budgets, and 2) propagating these priorities to the shared SearchAgent-X scheduler.

**Other Tools** The core ideas of SearchAgent-X, including balancing retrieval accuracy and system efficiency, priority-aware scheduling, and allocating "just enough" tool-use effort, can be applied broadly to any scenario with multi-round, dynamically autonomous tool invocations.

## E  THE USE OF LARGE LANGUAGE MODELS

We use the large language model for proofreading and language refinement for this paper. The LLM's contribution is confined to improving grammar and prose; all research concepts and results are the original work of the authors.

