# OpenReview forum: "Demystifying and Enhancing the Efficiency of Large Language Model Based Search Agents"
_ICLR.cc/2026/Conference — ICLR 2026 Poster_

### Official Review · Reviewer_pjDb · 2025-10-31

**Soundness:** 3
**Presentation:** 3
**Contribution:** 3
**Rating:** 6
**Confidence:** 3

**Summary:**

This paper addresses the significant system efficiency bottlenecks in LLM-based search agents, which operate by dynamically interleaving reasoning (generation) and retrieval. The authors first analyze the sources of inefficiency, demonstrating a non-monotonic relationship between retrieval accuracy and end-to-end efficiency. They find that both exact search (high retrieval overhead) and low-recall approximate search (high reasoning overhead) are suboptimal, showing high-recall approximate search is the best compromise.

Second, they diagnose a "cascading latency" problem in existing systems (like vLLM) caused by two primary factors: (1) Improper scheduling, where standard FCFS policies fail to optimize for search-agent workloads, leading to poor KV-cache utilization; and (2) Retrieval-induced stalls, where timing misalignments between asynchronous retrieval and generation steps force requests to wait, resulting in cache evictions and costly recomputation.

To address these challenges, the authors propose SearchAgent-X, a high-efficiency inference framework. SearchAgent-X introduces two main techniques: A priority-aware scheduler that replaces FCFS. It prioritizes requests based on context length, retrieval count, and waiting time to maximize KV-cache reuse. A non-stall retrieval mechanism, an adaptive early-termination strategy for ANN search. This strategy monitors retrieval "maturity" and synchronizes the search termination with the LLM engine's readiness, thus eliminating stalls.

Experiments show that SearchAgent-X significantly outperforms baselines, achieving up to 3.4x higher throughput and 5x lower latency while maintaining the generation quality of exact-retrieval systems.

**Strengths:**

- The paper tackles a critical and timely problem. As agentic systems that interact with external tools become more common, their inference efficiency is a major bottleneck. This is a strong systems-level contribution.
- The analysis in Section 2.2 is a key contribution. The identification of the non-monotonic efficiency curve (Figure 1) is a clear insight. The decomposition of latency amplification into "improper scheduling" and "retrieval-induced stalls" (Figure 2) is clear, well-illustrated, and clearly motivates the solution.
- The solutions are sound and well-motivated.
The priority-aware scheduler is a smart improvement over FCFS, as it explicitly models the value of a request's KV-cache prefix.
The non-stall retrieval mechanism is the most novel part. It's not just naive early stopping; it's a true co-design where the retrieval component's behavior is dynamically adapted based on the generation engine's state. This tight coupling is a key insight.
- The experiments are comprehensive and convincing.
The system is benchmarked against strong baselines (vLLM with ENN, vLLM with ANN, and vLLM+Cache+ANN).
The evaluation covers both offline and online serving scenarios, showing it's robust.

**Weaknesses:**

- The priority scheduling mechanism (Section 3.2) feels a bit heuristic. It relies on discretizing three metrics ($R_i, C_i, W_i$) into $G$ levels and assigning priority based on whether any metric exceeds a threshold (Eq. 2). The logic seems sensitive to the thresholds $T_{M,k}$, and the logic for combining these metrics isn't deeply explored.


- The non-stall retrieval mechanism (Section 3.3) depends on retrieval maturity and "the readiness of the LLM engine." The implementation of this "readiness" signal is critical but vague. Algorithm 1 (line 9-11) abstractly mentions CheckExternalNonStallSignal() when the LLM Engine has waiting requests. How is this practically implemented?

- The non-stall retrieval mechanism is tested on HNSW and NSW, which are iterative, graph-based ANN methods. The "retrieval maturity" concept is tied to this iterative process. It's unclear how this would apply to other major ANN algorithms (e.g., PQ, IVF) that aren't iterative in the same way. I am curious what are the search solution in real-world industry like perplexity

**Questions:**

see weakness

---

> ### Author Response · Authors · 2025-11-21
>
> We are grateful for the reviewer’s very positive evaluation and clear understanding of our goals. Below we clarify the remaining minor concerns.
>
> ---
>
> ### W1 – Are the priority rules too heuristic or sensitive?
>
> Our design is **deliberately mild and generic**, rather than brittle. We state on two perspectives as follows.
>
> - **Tha layered design** allows small differences within a layer and avoids unnecessary priority changes due to tiny metric variations.
>     - We do not fix absolute thresholds. Instead, we compute per-batch quantiles of three metrics (context length, retrieval count, waiting time) and use these quantiles to define priority layers.
>
>     Thus, the thresholds adapt dynamically to the current system state.
>
> - **When combining metrics**, our goal is to improve efficiency while avoiding starvation:
>     - if a request’s waiting time becomes large, its priority increases; if its reasoning has been stalled for many iterations, its priority also increases.
>     - Intuitively, these metrics act like legs for walking: each leg (i.e., each metric) cannot lag too far behind, otherwise the progress would be hindered.
>
> Sensitivity experiments (Figure 10 and 11) show stable improvements once \(G >= 6\), indicating that the design is robust while remaining simple.
>
> ---
>
> ### W2 – Implementation of the “readiness” / non-stall signal
>
> - The readiness signal is indeed central to non-stall retrieval. As described in Section 3.3 and Appendix B.2:
>     - as the search proceeds, each retrieval request exhibits a knee point where the marginal gain in recall per additional search step becomes negligible;
>     - this knee point defines retrieval maturity.
> - Traditional **fixed-step search** is inflexible: it cannot adapt to different request difficulties and may either under-search or over-search. Our non-stall retrieval instead:
>     - monitors the **marginal benefit** of new search steps,
>     - declares the request mature when the marginal gain falls below a threshold, and
>     - uses this maturity signal as the external non-stall signal in Algorithm 1.
> - In **Figure 8**, we show a case where, after the maturity point, recall already exceeds **0.98**. An ex-post validation in **Section C.3** and **Figure 9** demonstrates that: the maturity point accurately captures each request’s difficulty; retrieval cost is significantly reduced; and high recall is maintained.
>
> ---
>
> ### W3 – Other ANNs are also applicable, just with coarser granularity
>
> - We focus on graph-based ANN (HNSW/NSW) because they currently represent the best precision–latency trade-offs [1], and are widely adopted in modern vector databases and industrial systems.
> - However, **the non-stall principle is not limited to graph-based indices**:
>     - PQ- and IVF-based methods are also iterative in nature (e.g., over clusters or coarse partitions).
>     - These methods expose tunable iterative-manner parameters (e.g., the number of clusters probed) whose increase improves recall but also increases latency, similar to the search step in HNSW.
>     - Our framework can treat these iterations/clusters as units of search effort and apply the same maturity-point logic.
> - In large-scale industrial deployments (e.g., e-commerce or enterprise search), it is common to [2]:
>     - run HNSW- or IVF-based indices in multi-threaded, multi-node setups, and
>     - tune recall/latency trade-offs via parameters analogous to `efSearch` or `nprobe`.
> - SearchAgent-X is compatible with such systems as long as the retrieval engine exposes a monotone effort–recall trade-off curve, which these indices already provide.
> - We see exciting potential in applying SearchAgent-X-style non-stall retrieval and priority scheduling in production environments and are actively exploring this direction! Thanks very much for your pointing out! We `have added discussion in Section D.1 paragraph 2`.
>
> ---
>
> ### References
>
> [1] Wang, M., Xu, X., Yue, Q., & Wang, Y. (2021). A comprehensive survey and experimental comparison of graph-based approximate nearest neighbor search. *Proceedings of the VLDB Endowment, 14*(11), 1964–1978. https://doi.org/10.14778/3476249.3476255
>
> [2] Ren, Z., Doekemeijer, K., Apparao, P., & Trivedi, A. (2025, October). Storage-based approximate nearest neighbor search: What are the performance, cost, and I/O characteristics? In *2025 IEEE International Symposium on Workload Characterization (IISWC)*. IEEE.

---

> ### Author Response · Authors · 2025-11-29
> **Discussion Summary of Reviewer pjDb by Authors**
>
> Reviewer pjDb offered an enthusiastic assessment, highlighting our problem formulation, the clarity of our bottleneck analysis (particularly the non-monotonic efficiency curve and latency decomposition), and describing non-stall retrieval as "the most novel part" due to its true co-design of retrieval and generation. We appreciate this thoughtful analysis very much.
>
> The remaining concerns were minor and easily addressed: as a same concern with Reviewer 3XfA, we clarified that priority scheduling is intentionally simple yet robust—using dynamic per-batch quantiles rather than fixed thresholds—with sensitivity studies (Figures 10–11) confirming stable gains across configurations. For the "readiness signal" implementation, we pointed to the maturity-point mechanism detailed in Section 3.3 and Appendix B.2, with Figure 8 showing recall exceeding 0.98 at termination. On generalization beyond graph-based ANN, we explained that IVF and PQ methods similarly expose tunable effort–recall knobs (e.g., nprobe), making them fully compatible with our framework. This was a highly constructive review that validated our core contributions while helping articulate the implementation details and broader industrial applicability—both now strengthened in the revision.

---

### Official Review · Reviewer_eCQa · 2025-11-01

**Soundness:** 2
**Presentation:** 2
**Contribution:** 2
**Rating:** 4
**Confidence:** 2

**Summary:**

This paper presents SearchAgent-X, an inference framework for Large Language Model (LLM)-based search agents that incorporate priority-aware scheduling and non-stall retrieval. The design is based on the observation that both excessively high and excessively low retrieval accuracy degrade efficiency. SearchAgent-X improves the throughput by 1.3-3.4x

**Strengths:**

- The proposed method achieves good latency/throughput improvement while preserving the accuracy of search agents.

**Weaknesses:**

- No discussion about the previous efficiency optimization method for RAG with multiple rounds of retrievals, including [1-3]
- The non-stall early termination strategy in Section 3.3 seems inconsistent with the retrieval-induced stalls in Section 2.2.2. It is unclear how this optimization resolves the stall caused by being late for the LLM generation steps.
- The paper contains several unclear points. See the questions below.

### References

- [1] https://arxiv.org/abs/2502.20969
- [2] https://arxiv.org/abs/2503.14649
- [3] https://arxiv.org/abs/2505.07833

**Questions:**

- In Section 2.2, what model and dataset did you use, and what was the configuration of the inference engine (e.g., batch size)?
- In Figure 2a, I cannot see how the latency increases by 83x. I only observe the end-to-end latency rising from around 180s to 240s. Also, why are the data points so coarse-grained (e.g., no points between retrieval latencies of 1s and 4s)?
- In Figure 2b, what is d_i​? How is the scheduling performed here (is it rescheduled after each retrieval step)? Also, how can a non-FCFS policy solve this issue? For example, even if we prioritize request a over request b, wouldn’t that still lead to a cache miss for request b?
- In Figure 2c, what is the latency overhead caused by retrieval-induced stalls? If scheduling is done at the iteration level and chunked prefill is used, I think this overhead should be small.
- Section 4.1 mentions using two models from Search-R1 and ReCall, but the figures do not specify which model was used.
- In Section 4.3, it compares the advantages of prefix caching for top-k=5 and top-k=1, but where is the data for top-k=1?
- Section 4.3 mentions that non-stall retrieval reduced end-to-end latency by 41s, but it is unclear which data this refers to.
- It seems that most of the performance benefit comes from priority scheduling, and I think it is not limited to search-based agents. I wonder if this technique can be generalized to other types of LLM agents that use external tools.

---

> ### Author Response · Authors · 2025-11-21
>
> We thank the reviewer for recognizing the latency/throughput gains of SearchAgent-X. We believe that several minor concerns arise from misunderstandings, which we address point-by-point below.
>
> ---
>
> ### W1 – Previous work on multi-round RAG system does not address **inefficiency in the dynamic search–reasoning interleaving scenario**
>
> - To our knowledge, this is the first system optimization work explicitly targeting dynamic search–reasoning interleaving in search agents, i.e., where retrieval and generation are tightly interwoven at runtime.
>     - TELERAG primarily prefetches retrieval results to reduce the cost of the batched retrieval stage, rather than optimizing the behavior of an end-to-end interactive search agent system.
>     - RAGO and PATCHWORK focus on adjusting retrieval/generation knobs (e.g., sampling or retrieval parameters), not on analyzing or redesiging the serving system for dynamic tool-using agents.
>     - All these works, and other system-level RAG efforts (e.g., RAGServe, RAGCache, PipeRAG) discussed in the related work section target different aspects of RAG pipelines but **do not analyze the root causes of system inefficiency in the dynamic search–reasoning interleaving scenario like SearchAgent-X**.
> - Thanks for your mentioning. We `have added this clarification in the revision (see Section 5 paragraph 1)`.
>
> ---
>
> ### W2 – Non-stall retrieval makes **“just-right letting go of retrieval” to address retrieval-generation misalignments**
>
> - We’d like to clarify that the design logic is consistent with motivations — even small timing misalignments between retrieval completion and LLM token generation can cause:
>     - idle waiting (stalls) and
>     - cache evictions leading to expensive recomputation (See response to Q4 below for more details about retrieval-induced stalls).
> - Non-stall retrieval thus performs **“just-right letting go of retrieval”**: we stop retrieval as soon as it is mature enough for the current generation step, instead of insisting on maximal recall regardless of the LLM engine’s state.
> - In other words, we align the termination of retrieval with the readiness of the generation engine, thereby directly addressing the retrieval-induced stalls diagnosed in Section 2.2. Please refer to the answer of Q4 for empirical evidence of stalls.
>
> ---
>
> ### W3 – Clarifications on unclear points
>
> Below we respond to each specific question and will incorporate these clarifications into the revised text.
>
> ---
>
> ### Q1 – Experiment setup of motivations has been described more explicitly
>
> - **Model and dataset:** Search-R1 search agent on the MuSiQue dataset.
> - **Inference engine configuration:** vLLM with its default batch size and PagedAttention configuration.
> - These choices ensure that the observed latency amplification effects are representative of realistic search-agent workloads.
> - Thanks for the reminder! We `have added descriptions in Section 2.2 paragraph 1`.
>
> ---
>
> ### Q2 – Interpreting the 83× latency amplification and data points
>
> - The 83× factor denotes that, **compared to a traditional sequential RAG pipeline**, the same minor increase in retrieval latency can cause up to an 83× larger increase in end-to-end latency for a search agent.
> - In Figure 2a, each point corresponds to a different setting of ANN parameters (e.g., different `efSearch` values). The retrieval latencies on the x-axis are thus **not manually chosen grid points, but the natural results of different retrieval parameter configurations**.
> - We `have explicitly stated how these points are obtained (see Section 2.2.2 paragraph 3)`.
>
> ---
>
> ### Q3 – Meaning of \(d_i\), scheduling policy, and cache misses
>
> - In Figure 2b, \(d_i\) denotes the decoding step index. Each LLM token generation step is either a prefill or a decoding step.
> - Retrieval and generation are executed fully asynchronously to maximize efficiency. This asynchrony is precisely what creates the misalignment leading to stalls and cache evictions.
> - The aim of the figure is not to show that a non-FCFS policy can prevent *all* cache misses for all requests. Instead:
>     - request `a` has a more valuable KV prefix and is closer to completion; a cache miss on `a` is more costly than on `b`,
>     - so a priority-aware policy chooses to serve `a` first, keeping its cache warm and allowing it to complete sooner.
>     - Once `a` finishes, it frees GPU memory and KV-cache capacity, which ultimately benefits `b` as well. The policy therefore optimizes for the relative value of cache misses, not just their count.

---

> ### Author Response · Authors · 2025-11-21
>
> ---
>
> ### Q4 – Retrieval-induced stalls leads to 25% of sequences that could have been wisely scheduled, thus loosing the opportunity of further 6.31% latency improvement
>
> - The problem is not the computational cost of retrieval itself, but that valuable KV-cache entries are evicted while retrieval lags behind generation.
> - As reported, on average **more than 25%** of sequences experience such retrieval-induced stalls that could have been avoided across various scenarios.
> - After applying non-stall retrieval on top of other optimizations, we further reduce end-to-end latency by **6.31%** (as seen in Figure 6 left). We also observe **a** **naive early stop** (without aligning with LLM generation steps) **shows obviously lower efficiency** (see Section C.6 and Table 7).
>
> ---
>
> ### Q5 – Main experiments use SearchAgent-X model (another model in appendix), which has been stated more clearly
>
> - The experiments of Section 4.1 focus on Search-R1, which is both highly effective and widely adopted, making it a representative search agent.
> - Additional results for other models, search methods, and datasets are provided in Appendix C, where we consistently observe that SearchAgent-X outperforms the baselines.
> - `Added descriptions can be seen in Section 4.2 & 4.3 paragraph 1`.
>
> ---
>
> ### Q6 – Explaining top-k data in Section 4.3
>
> We show results on top-k=1,2,3,4,5 for comparing overall performance in Section 4.2. For ablation in Section 4.3, we focus on top-k=5, because we need to choose a subset, and top-k=5 makes the most sense. In Section 4.3, we mention topk=1 just to make a point about task difficulties, but do not intend to compare all top-k for ablations.
>
> - The ablation in Section 4.3 focuses on top-k = 5 because it represents the most challenging scenario, with:
>     - longer generation lengths,
>     - more retrieval rounds, and
>     - stronger resource contention.
> - This setting makes the differences between techniques more pronounced. However, to observe task difficulties, we also analyze how much prefix caching itself can help and find:
>     - the relative advantage of prefix caching decreases with larger top-k,
>     - which is intuitive since larger top-k leads to longer trajectories and heavier contention. That’s also why SearchAgent-X optimizations are essential.
> - The prefix caching improvement data for all top-k values is actually **already shown in Section 4.2 Figure 4**. We `have cross-cited and explained this point in Section 4.3`. Thanks!
>
> ---
>
> ### Q7 – Interpreting the 41-second latency reduction
>
> - The reported 41-second reduction refers to the end-to-end latency shown in **Figure 6 (left)**, aggregating over full request trajectories.
> - **Figure 6 (right)** illustrates the timeline for a single sequence, which is only one component of a multi-sequence request.
> - A single user request may involve multiple sequences, so the end-to-end improvement accumulates across sequences; we `have added clarifications for this mapping in Section 4.3 paragraph 2`.
>
> ---
>
> ### Q8 – Generalizing beyond search-based agents is absolutely feasible!
>
> - We appreciate this important insight. The core ideas of SearchAgent-X:
>     - balancing retrieval accuracy and system efficiency,
>     - priority-aware scheduling, and
>     - allocating “just enough” tool-use effort
>     can be applied broadly to any scenario with **multi-round, dynamically autonomous tool invocations**.
> - We have indeed been exploring these **exciting principles in** **RL post-training frameworks** **with tool use**, aiming at releasing open-source repos.
> - This broader applicability `has been explicitly highlighted (in Section D.2 paragraph 2)`! Thank you!

---

> > ### Comment · Reviewer_eCQa · 2025-11-27
> >
> > Thanks for your response. I have some follow-up questions.
> >
> > ### Q2
> > I still don't understand how you computed the numbers like 83x or 17x in the figure. Could you provide concrete numbers and explain how they correspond to the figure? In the current state, the figure is quite misleading. Also, please explain the meaning of the `efSearch` parameter and what value you set for that.
> >
> > ### Q3
> > How can a cache miss on some request be more costly than another request? Does the recomputation cost not solely depend on the number of tokens?

---

> > > ### Author Response · Authors · 2025-11-29
> > >
> > > Thanks very much for your timely feedback!
> > >
> > > ## Q2: Concrete numbers/settings for the retrieval latency figure.
> > >
> > > **Original Data**. Below we present our original data of Figure 2(a). The second column presents a modular sequential RAG where one time of retrieval is done strictly before generation. The third column presents search agents. We calculate the increase ratio (on top of the search range = 500) of the third to the second to show their change trend, as shown in the fourth column - which is also how “magnification” in Figure 2(a) is calculated.
> > >
> > > The results confirm that a little bit increase of retrieval latency, though not severely affecting modular RAG efficiency, could harm search agent end-to-end efficiency. Thanks for your reminder of figure interpretation. `We have added more detailed explanation in Figure 2 caption`.
> > >
> > > | ANN Search Range | Avg. Retrieval Latency (Modular Sequential RAG) | Avg. End-to-End Latency (Agentic RAG e.g., Search-R1) | Ratio of Increase by Base | Prefix Hit Rate |
> > > | --- | --- | --- | --- | --- |
> > > | 500 (base) | 0.6106 | 177 |  | 0.3082 |
> > > | 5000 | 0.6586 (+0.048 compared with base) | 178 (+1 compared with base) |  | 0.3057 |
> > > | 10000 | 0.8151 (+0.204 compared with base) | 194 (+17 compared with base) | **17/0.2044 = 83** | 0.2816 |
> > > | 20000 | 4.3622 (+3.752 compared with base) | 239 (+62 compared with base) | **62/3.75 = 17** | 0.2046 |
> > >
> > > **How data points are selected with parameter efSearch**. the efSearch knob is typically what people use to balance retrieval accuracy vs. latency in graph-based search. Hence, we do not simulate retrieval with fixed time ranges; rather, we control efSearch to see how modular sequential RAG and search agents perform, for approaching practical use cases as much as possible. Since retrieval latency does not linearly increase with search range efSearch, the X values of points in Figure 2(a) are not evenly distributed.
> > >
> > > ## Q3: Not just token number matters, timing also matters
> > >
> > > As an inference system serves many requests dynamically, scheduling should consider more than static context length. That said, SearchAgent-X could benefit from prioritizing a request (say, r1) rather than another request (say, r2) from three aspects: (1) r1 has a longer context (token number), so recomputing r1 is more expensive; (2) r1 is likely to complete inference with fewer tokens, so GPU resources can be released sooner; (3) r1 has waited for a longer time, so completing r1 helps increase fairness.
> > >
> > > These factors are precisely what motivates our design of the priority-aware scheduler.

---

> > > > ### Author Response · Authors · 2025-11-29
> > > > **Discussion Summary of Reviewer eCQa by Authors**
> > > >
> > > > This discussion with Reviewer eCQa was the most detailed, involving a productive follow-up round that allowed us to clarify several technical misunderstandings. The reviewer's initial concerns centered on missing comparisons to prior multi-round RAG systems and confusion about key figures and metrics. We explained that existing works (TELERAG, RAGO, PATCHWORK) target fundamentally different aspects of RAG pipelines—none analyze the dynamic search–reasoning interleaving scenario that SearchAgent-X addresses. The most substantive back-and-forth concerned Figure 2a's "83× latency magnification": in response to the reviewer's request for concrete numbers, we provided a detailed data table showing how minor retrieval latency increases (e.g., +0.2s) translate to disproportionately large end-to-end slowdowns (+17s) in search agents versus modular RAG—yielding the reported 83× ratio. We also clarified that cache miss "cost" depends not just on token count but on timing factors (proximity to completion, waiting time, resource release), which is precisely why priority-aware scheduling outperforms naive FCFS.
> > > >
> > > > The reviewer's questions ultimately helped us strengthen the paper's exposition, and we have incorporated these clarifications—including explicit experimental configurations and figure annotations—into the revision. It also excites us that the reviewer saw the potential of our design beyond the search agent scenario.

---

### Official Review · Reviewer_PMHy · 2025-11-01

**Soundness:** 3
**Presentation:** 3
**Contribution:** 3
**Rating:** 6
**Confidence:** 3

**Summary:**

This paper systematically investigates the efficiency bottlenecks of LLM-based search agents and uncovers several insights: (1) retrieval accuracy exhibits a non-monotonic relationship with end-to-end efficiency; (2) search agents are extremely sensitive to retrieval latency, where minor delays can lead to an 83× latency magnification effect; and (3) traditional FCFS scheduling results in low KV-cache utilization. Based on these findings, the paper proposes SearchAgent-X, a system that optimizes search agents through high-recall approximate retrieval, priority-aware scheduling, and a non-stall retrieval mechanism. Experiments demonstrate that the system achieves up to 3.4× higher throughput and 5× lower latency while maintaining generation quality.

**Strengths:**

1. With the rise of DeepResearch-related work, the paper systematically studies efficiency issues in this emerging scenario, offering significant practical significance and application value.
2. Through empirical studies, the paper discovers many noteworthy phenomena. It provides a deep analysis of the root causes of search agent latency, supported by extensive experiments.
3. Priority scheduling avoids complex weight tuning through discretization, while non-stall retrieval adaptively terminates based on retrieval maturity. Both techniques are easy to implement and model-agnostic.

**Weaknesses:**

1. The main improvements stem from the high reusability of KV-cache, which works for the Search-R1 pattern. However, some current search agents modify historical context to handle overly long retrieval trajectories (e.g., summarizing previous context or keeping only recent k rounds of retrieval results), which could render the priority-aware scheduling completely ineffective.
2. Priority-aware scheduling cannot be applied to commercial models (e.g., GPT), and the non-stall retrieval mechanism cannot be used to retrieve APIs (e.g., Google). These limitations make the application scenarios quite restricted (local LLM, knowledge base retrieval, KV-cache constrained, Search-R1 pattern).
3. Lack of generalization discussion across different knowledge bases and tasks. For more challenging tasks or models that require more retrieval iterations, many designs in this paper—such as concurrency levels, priority level G, and maturity thresholds—may require substantial adjustments.

**Questions:**

1. The paper mentions that efSearch=500 may lead to unstable recall. What is the specific performance degradation across different benchmarks?
2. How does SearchAgent-X perform on tasks of varying difficulty? What are the latency and throughput improvements over baselines across different tasks?

---

> ### Author Response · Authors · 2025-11-21
>
> We thank the reviewer for their high recognition of our study in emerging DeepResearch-style intelligent agents, as well as the simplicity and generality of our design.
>
> ---
>
> ### W1 – The scheduler still works as long as prefix caching is used
>
> - Techniques such as context trimming, summarization, or token sparsification are a complementary line of work that reduce the effective context length to improve efficiency. This methods don't make prefix caching unavailable, as long as sparsification is applied in a consistent, prefix-preserving manner during decoding.
> - In this case, our scheduler can exploit real-time available prefix length as a scheduling metric.
>
> ---
>
> ### W2 – Applicability to commercial models and non-local retrieval
>
> - **Priority-aware scheduling have already been used in commercial settings.**
>     - Priority- or SLO-aware schedulers have already been explored in production LLM serving systems, where non-FCFS scheduling significantly improves throughput and tail latency for heterogeneous workloads [1].
>     - For complex tasks such as DeepResearch-style agents, users naturally expect longer waiting times, making backend priority scheduling particularly attractive: it can reduce serving cost while keeping user-visible latency reasonable.
>     - Our design directly targets such heterogeneous, long-running agent workloads, and the same ideas can be applied to proprietary models exposed via APIs, where scheduling happens on the user’s serving stack.
> - **Our experiments assume local retrieval, but insights could be widely applicable.**
>     - We would like to clarify that local retrieval has already become an essential part in modern LLM agents [2], which is why we explore its optimizations.
>     - While non-stall retrieval assumes controlable searches, it is still possible to initiate concurrent requests to search APIs with multiple different search parameters, and do non-stall retrieval with an asynchronous manner as our compatible extension.
>     - Besides, the underlying insight is more general: many vector search systems and APIs expose knobs that trade recall against latency (e.g., search effort parameters analogous to `efSearch` or `nprobe`) [3]. Our analysis—that both excessively high and excessively low retrieval accuracy degrade efficiency—thus applies whenever such recall–latency trade-off knobs are available, including remote search APIs.
> - Thanks for the valuable reminder! We `have added relevant discussions about local retrieval in Section D.1 paragraph 3`.
>
> ---
>
> ### W3 – Scalability has been extensively verified; larger production scale is expected to yield consistent improvements
>
> - Our experiments already cover scalable:
>     - model sizes and top-k values (Table 2),
>     - search models (Table 4),
>     - retrieval schemes (Table 8),
>     - task datasets (Table 3), and
>     - concurrency controls (Table 6),
>     and consistently show substantial throughput and latency improvements over strong baselines.
> - SearchAgent-X is fundamentally a system-level design with minimal assumptions about the knowledge base content or task distribution; it only requires that:
>     - search is called multiple rounds for each request
>     - search effort can be tuned (e.g., ANN parameters), and
>     - generation is scheduled at iteration/token granularity.
> - Combing our observations on difference task difficulties (see Q2 below), we are convinced that our system-level optimizations could provide consistent great benefits when scaling up to real production scenartios.
>
> ---
>
> ### Q1 – efSearch = 500 leads to significantly lower query recall and is therefore not suitable for search agents
>
> - In our previous analysis, efSearch = 500 ****was shown to harm end-to-end system (max. of **20% lower p99 recall** compared with ideal retrieval in Musique dataset) for two reasons:
>     - **Query difficulty heterogeneity.** Queries vary significantly in difficulty. For a subset of hard queries, a fixed search range (e.g., 500) is simply insufficient, which leads to degraded recall specifically on the tail queries. This is exactly where high-quality retrieval matters most for overall robustness.
>     - **Under-utilized time budget.** When the retrieval finishes early at efSearch = 500 but vLLM has not yet advanced to the next generation step, the retrieval thread becomes idle. This wastes wall-clock time that could have been used for additional search steps. In contrast, using a higher efSearch combined with our early-stop (maturity-based) criterion allows the system to fully exploit the available time budget: when there is slack, we can search more thoroughly; when the LLM is ready, we stop as soon as the retrieval is mature.
> - To further address your concern, we are running end-to-end experiments to show exact benchmark metric comparisons. We will update the paper as soon as completed.

---

> > ### Author Response · Authors · 2025-11-21
> >
> > ---
> >
> > ### Q2 – SearchAgent-X demonstrates better improvements on more difficult tasks where optimization is more critical
> >
> > - Empirically, we observe that **SearchAgent-X often yields larger gains on harder tasks**. As shown in Table 3, the HotpotQA dataset (a easier dataset according to the Search-R1 paper) yields lower benefits compared to the Musique dataset in our main experiments (e.g., **1.52x** of throughput improvement on easy dataset vs. **1.84x** of throughput improvement on hard dataset, for top-k=3).
> > - This is expected because harder tasks typically involve:
> >     - more search iterations,
> >     - longer reasoning trajectories,
> >     - more complex system states,
> >     - denser retrieval/token generation patterns, and
> >     - more intense resource contention.
> > - These factors amplify the impact of scheduling and retrieval–generation alignment, so **priority-aware scheduling and non-stall retrieval become especially important** in such settings.
> > - We `have highlighted this trend and explicitly point to the corresponding table/figure in the revision (see Section C.1 paragraph "Different Datasets")`.
> >
> > ---
> >
> > ### References
> >
> > [1] Sun B., Huang Z., Zhao H., Xiao W., Zhang X., Li Y., & Lin W. (2024). *Llumnix: Dynamic Scheduling for Large Language Model Serving*. In 18th USENIX Symposium on Operating Systems Design and Implementation (OSDI ’24), Santa Clara, CA, USA. USENIX Association.
> >
> > [2] Gao, Y., Xiong, Y., Gao, X., Jia, K., Pan, J., & Wang, H. (2023). *Retrieval-Augmented Generation for Large Language Models: A Survey*. arXiv preprint arXiv:2312.10997.
> >
> > [3] https://milvus.io/ai-quick-reference/why-are-high-recall-values-important-when-benchmarking-approximate-nearest-neighbor-searches-and-how-do-vector-databases-typically-trade-off-recall-for-speed?utm_source=chatgpt.com

---

> > > ### Author Response · Authors · 2025-11-29
> > > **Discussion Summary of Reviewer PMHy by Authors**
> > >
> > > We thank the reviewer for the thoughtful question about the applicability of SearchAgent-X beyond the specific Search-R1 pattern—particularly regarding agents that modify historical context, commercial API-based models, and generalization across diverse tasks. Our response clarified that (1) our scheduler remains effective whenever prefix caching is available, even under context trimming or summarization; (2) priority-aware scheduling is already deployed in production LLM serving systems and the core insight—that recall–latency tradeoffs exist in most retrieval systems—extends naturally to remote APIs with tunable search parameters.
> > >
> > > On generalization, we emphasized that our experiments already span multiple model sizes, retrieval schemes, datasets, and concurrency levels, consistently yielding strong improvements. Notably, we highlighted an encouraging trend: SearchAgent-X delivers larger gains on harder tasks (e.g., 1.84× throughput on Musique vs. 1.52× on HotpotQA), precisely because difficult queries amplify resource contention where our optimizations matter most. We appreciate the reviewer pushing us to elaborate the broader applicability of our work, and have incorporated these clarifications into the revision.

---

### Official Review · Reviewer_3XfA · 2025-11-01

**Soundness:** 3
**Presentation:** 3
**Contribution:** 2
**Rating:** 6
**Confidence:** 4

**Summary:**

This paper investigates efficiency bottlenecks in LLM-based search agents, which dynamically interleave reasoning and retrieval during generation. The authors identify that both exact and coarse retrieval hurt performance and the framework is highly sensitive to retrieval latency. To address this, the authors propose SearchAgent-X, which features high-recall approximate retrieval, priority-aware scheduling and non-stall retrieval.

**Strengths:**

- The paper is well-written and easy to follow.
- The paper shows the non-monotonic relationship between retrieval accuracy and efficiency.
- Components (priority scheduling, non-stall retrieval) in SearchAgent-X are well-motivated.
- Comprehensive ablation studies demonstrate robustness of the proposed method.

**Weaknesses:**

- The baselines do not include recent LLM serving framework like SGLang.
- The priority discretization thresholds and non-stall retrieval maturity criterion involve heuristic settings without extensive sensitivity analysis.

**Questions:**

- Can the proposed non-stall retrieval mechanism degrade answer quality when retrieval terminates too early?
- Could priority scheduling introduce starvation or fairness issues under high request concurrency?
- How would the framework adapt to multi-agent environments, where multiple reasoning agents interact concurrently?

---

> ### Author Response · Authors · 2025-11-21
>
> We thank the reviewer for the very positive assessment of our motivation, analysis, and ablation studies! Below we address the remaining light concerns.
>
> ---
>
> ### W1 – Other frameworks like SGLang are also applicable
>
> - Our implementation of **SearchAgent-X is a direct modification of vLLM**. We therefore compare against vLLM-based baselines so that all systems share the same serving stack (PagedAttention, batching logic, etc.), avoiding interference with the performance observations from unrelated optimizations.
> - However, The proposed optimization techniques are **not specific to vLLM**. SearchAgent-X only assumes:
>     - token-level generation scheduling, and
>     - an FCFS (or FCFS-like) base policy that can be overridden.
> - SGLang also adopts token-level generation scheduling and an FCFS-style policy, **so SearchAgent-X can be integrated into SGLang with minimal changes for system improvement**. We `have added relevant explanations in Section D.1 paragraph 1`.
>
> ---
>
> ### W2 – Sensitivity analysis shown in Appendix; all designs are deliberately mild and generic
>
> - **Priority thresholds.**
>     - Please refer to the sensitivity study over the number of priority levels \(G\) (**Figure 10 and 11**). In our case, as long as the granularity is greater than 6, the throughput/latency improvements are consistently stable.
>     - This robustness stems from a **mild, layered scheduling design**: we assign priorities based on real-time **quantiles** of three metrics (context length, retrieval count, waiting time), rather than a hand-tuned weighted sum or fixed thresholds that require prior knowledge of workloads.
> - **Retrieval maturity / non-stall termination.**
>     - SearchAgent-X uses a lightweight offline profiling run to estimate the gradient of retrieval benefit per additional search step for each query. From this, we identify **a safe gradient signal** where marginal recall gain becomes negligible.
>     - This knee point is directly visualized in Figure 8, and an ex-post evaluation in **Figure 9** shows that terminating at this gradient maintains high recall while substantially reducing retrieval effort. We also observe a naive early stop (without aligning with LLM generation steps) shows obviously lower efficiency (see **Section C.6 and Table 7**).
> - **Key takeaway.** All designs in SearchAgent-X are deliberately simple and direct—hence generic—yet already deliver strong gains, which highlights the core empirical observations in our motivation.
>
> ---
>
> ### Q1 – Non-stall retrieval does not degrade answer quality
>
> - The end-to-end generation quality in **Table 1 and 5** shows our non-stall retrieval does not sacrifice generated answer quality (compared with the exact search).
> - This robustness comes from designing non-stall retrieval **per request based on its search benefit gradient**, rather than stopping at a fixed number of search steps for all queries.
>
> ---
>
> ### Q2 – Starvation / fairness is considered and tackled
>
> - The priority scheduler explicitly accounts for both:
>     - each request’s arrival time, and
>     - its waiting time in the current round.
> - Our layered design ensures that if a request’s waiting time becomes dominant, it is guaranteed to be assigned the highest priority in subsequent iterations.
> - Consequently, long-waiting requests are promoted and served, effectively preventing starvation and fairness issues even under high request concurrency.
>
> ---
>
> ### Q3 – Adaptation to multi-agent environments
>
> - We agree this is a **very exciting and valuable direction!** In a multi-agent setup where multiple reasoning agents interact concurrently, resource contention becomes even more challenging.
> - Our priority-aware scheduling and non-stall retrieval mechanisms offer a natural starting point for coordinating across agents, for example by:
>     - introducing agent-level priorities or budgets, and
>     - propagating these priorities to the shared SearchAgent-X scheduler.
> - We consider this an important avenue for future work and `have discussed (in Section D.2 paragraph 1) multi-agent extensions in the revised paper`.

---

> > ### Author Response · Authors · 2025-11-29
> > **Discussion Summary of Reviewer 3XfA by Authors**
> >
> > We thank Reviewer 3XfA for recognizing our well-motivated analysis and comprehensive ablations. The discussion centered on two main concerns: (1) the absence of SGLang as a baseline, and (2) sensitivity of heuristic design choices. We clarified that SearchAgent-X is framework-agnostic—requiring only token-level scheduling and an overridable FCFS policy—and thus portable to SGLang with minimal effort; our vLLM-based comparison ensures a controlled evaluation isolating our contributions. For sensitivity, we directed the reviewer to existing appendix studies (Figures 10–11) demonstrating stable gains across a range of priority granularities, and explained that our designs are deliberately simple and generic, relying on real-time quantiles and lightweight offline profiling rather than workload-specific tuning. We also addressed quality and fairness concerns by pointing to end-to-end results (Tables 1, 5) confirming no degradation from non-stall retrieval, and detailing how our layered scheduler inherently prevents starvation. Overall, this was a constructive discussion that helped improve the presentation and interpretation of our approach.

---

### Meta-Review · Area_Chair_z8kf · 2026-01-06

**Summary:**

This paper systematically analyzes efficiency bottlenecks in LLM-based search agents and proposes SearchAgent-X, a framework employing high-recall approximate retrieval, priority-aware scheduling, and non-stall retrieval. It demonstrates significant throughput and latency improvements while preserving answer quality.

**Reviewer Concerns:**

The rebuttal effectively addressed most technical concerns and clarified misunderstandings. While some weaknesses regarding broader applicability and comparisons remain, they do not undermine the core contribution. The reviews converged on a "marginally above acceptance" rating.

**Reviewer Scores:**

I don't think these reviews will be significantly changed.

---

### Decision · Program_Chairs · 2026-01-26

Accept (Poster)